# A comprehensive AI model development framework for consistent Gleason grading
Xinmi Huo [1,24], Kok Haur Ong [1,24], Kah Weng Lau[2,3], Laurent Gole[2], David M. Young[2,4], Char Loo Tan [3], Xiaohui Zhu[5,6], Chongchong Zhang[7], Yonghui Zhang[7], Longjie Li[1], Hao Han [2], Haoda Lu[1,8], Jing Zhang[9], Jun Hou[10], Huanfen Zhao[11], Hualei Gan[12], Lijuan Yin[13], Xingxing Wang[10], Xiaoyue Chen[11], Hong Lv[12], Haotian Cao[9], Xiaozhen Yu[10], Yabin Shi[11], Ziling Huang[12], Gabriel Marini[1], Jun Xu [8], Bingxian Liu[14], Bingxian Chen[14], Qiang Wang[14], Kun Gui[14], Wenzhao Shi[15], Yingying Sun[16], Wanyuan Chen[16,17,18], Dalong Cao[19,20,21], Stephan J. Sanders [4,22], Hwee Kuan Lee[1], Susan Swee-Shan Hue [2,3,25] ✉, Weimiao Yu [1,2,8,25] ✉ & Soo Yong Tan [23,25] ✉

## Abstract

**Background** Artificial Intelligence(AI)-based solutions for Gleason grading hold promise for pathologists, while image quality inconsistency, continuous data integration needs, and limited generalizability hinder their adoption and scalability.

**Methods** We present a comprehensive digital pathology workflow for AI-assisted Gleason grading. It incorporates A!MagQC (image quality control), A!HistoClouds (cloud-based annotation), Pathologist-AI Interaction (PAI) for continuous model improvement, Trained on Akoya-scanned images only, the model utilizes color augmentation and image appearance migration to address scanner variations. We evaluate it on Whole Slide Images (WSI) from another five scanners and conduct validations with pathologists to assess AI efficacy and PAI.

**Results** Our model achieves an average F1 score of 0.80 on annotations and 0.71 Quadratic Weighted Kappa on WSIs for Akoya-scanned images. Applying our generalization solution increases the average F1 score for Gleason pattern detection from 0.73 to 0.88 on images from other scanners. The model accelerates Gleason scoring time by 43% while maintaining accuracy. Additionally, PAI improve annotation efficiency by 2.5 times and led to further improvements in model performance.

**Conclusions** This pipeline represents a notable advancement in AI-assisted Gleason grading for improved consistency, accuracy, and efficiency. Unlike previous methods limited by scanner specificity, our model achieves outstanding performance across diverse scanners. This improvement paves the way for its seamless integration into clinical workflows.

## Plain language summary

Gleason grading is a well-accepted diagnostic standard to assess the severity of prostate cancer in patients' tissue samples, based on how abnormal the cells in their prostate tumor look under a microscope. This process can be complex and time-consuming. We explore how artificial intelligence (AI) can help pathologists perform Gleason grading more efficiently and consistently. We build an AI-based system which automatically checks image quality, standardizes the appearance of images from different equipment, learns from pathologists' feedback, and constantly improves model performance. Testing shows that our approach achieves consistent results across different equipment and improves efficiency of the grading process. With further testing and implementation in the clinic, our approach could potentially improve prostate cancer diagnosis and management.

Prostate cancer (PCa) is a prevalent cancer among males, accounting for 15.1% of all male cancers diagnosed in 2020[1]. The Gleason Grade (GG) is a critical factor in assessing its aggressiveness and guiding treatment decisions. Pathologists evaluate tissue samples obtained from prostatectomy and recurrent/routine core biopsy to detect malignancies and stratify patients based on microscopic and histological characteristics. This diagnostic process is generally time-consuming and sometimes subjective, leading to potential misdiagnosis and over-diagnosis. To address these challenges, a technological solution is necessary to improve efficiency, accuracy, and consistency.

Recent advances in histology scanning technology and Artificial Intelligence (AI) offer great opportunities to improve the accuracy of GG. As

---

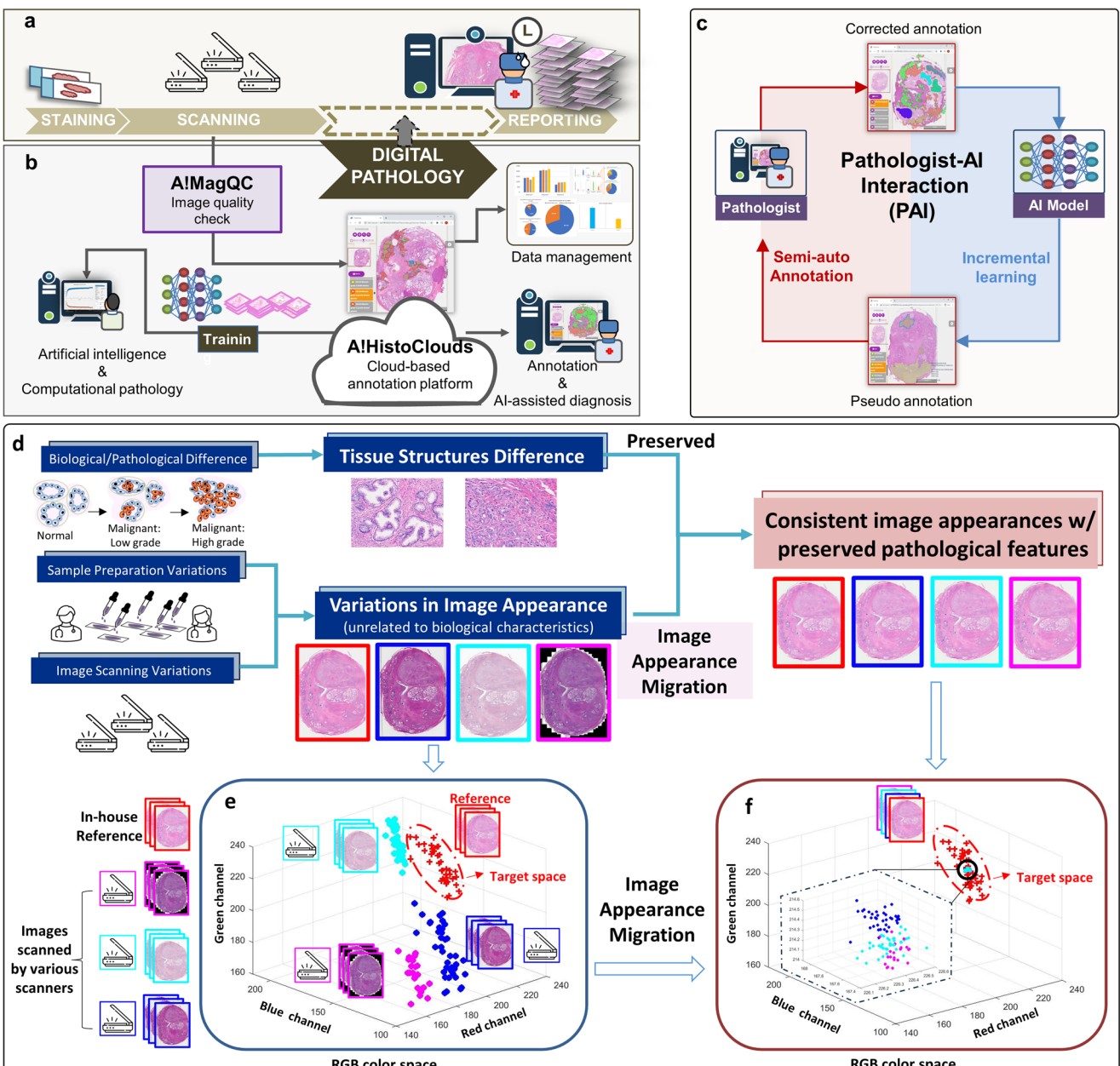

**Fig. 1 | Experimental design and pipeline overview.** The existing Digital Pathology assessment pipeline is illustrated in (**a**). In this study, we developed an integrated and comprehensive Digital Pathology image analysis pipeline, which is powered by A! MagQC (image quality assessment software), A!HistoClouds (Digital Pathology image viewer, annotation platform, and database), and an Artificial Intelligence model that can detect and grade prostate cancer for images scanned by multiple scanners, as shown in (**b**). The workflow of the pathologist-AI interaction (PAI) is presented in (**c**). After a base AI model is trained, it is applied to new data and generate pseudo annotations on Whole Slide Images. Pathologists review and modify the pseudo annotations using A!HistoClouds, and the corrected annotations are fed back to the model for further training. This process is repeated until the model achieves high accuracy. After meeting criteria, the model can generate accurate outcomes and assist pathologists in clinical diagnosis. In AI digital pathology, AI model performance is significantly influenced by image appearance variations. **d** Biological processes cause tissue structure differences, while sample preparation and scanning introduce notable appearance variations. Preserving pathological tissue structural differences is crucial for capturing biological features in AI learning for the purpose of diagnosis, but it's important to minimize appearance variations that don't relate to biological traits. Image appearance migration addresses this by migrating images from different scanners into a standard space. **e** Initially, mean RGB value distributions from various scanners may cluster distinctly due to appearance differences. **f** After applying image appearance migration, the data points converge into a compact, unified appearance, demonstrating the method's effectiveness in standardizing digital pathology images for AI analysis.

shown in Fig. 1a, the advent of glass-slide scanners has enabled the rise of Digital Pathology (DP) and the tantalizing promise of computational assessment. Whole Slide Image (WSI) has considerably expanded the volume of DP data by enabling digitalization of physical slides at high resolution. Meanwhile, our development of AI diagnostic models as shown in Fig. 1b, hold the potential to improve the efficiency and accuracy of pathological assessment by reducing turnaround time and enhancing detection consistency[2–5].

Both traditional Machine Learning(ML) and Deep Learning (DL) approaches have been widely explored for the detection, segmentation, classification of prostate tissues for the clinical assessment of Gleason scores[3–25]. Paige Prostate Alpha, an AI-based software system for PCa

assessment has been classified as a Class II device by the U.S. Food and Drug Administration (FDA), being the first FDA-approved AI product in the field of DP, but limited to the approved scanner. Independent studies have demonstrated that the utilization of such AI-assistance in PCa detection and grading has led to increased sensitivity and reduced diagnostic time[20–25].

Despite the recent achievements and developments of AI-based diagnostic solutions, the overall adoption of the AI models in pathology domain remains poor. Lack of comprehensive and systematically designed research and development pipeline on AI digital pathological diagnosis models is probably one of the main reasons. Many published studies suffer from several limitations. First of all, the dataset does not contain sufficient variations. For instance, many studies utilized only 1-2 scanners without incorporating a generalization solution into the pipeline[11–25]. The variation and quality of images acquired from different scanners, compounded by the variations of sample preparation and staining, might profoundly affect AI models[26], often leading to poor generalization during onsite testing and deployment. Second, the annotation schemes in many studies are often not well designed and inefficient to directly address the diagnostic needs. Large quantities of high-quality annotated data are essential for training DL models[5], but acquiring such annotations is expensive and time-consuming, requiring experienced pathologists to manually perform detailed annotations. Finally, the feasibility of evolving the existing AI-based models with more data has not been adequately considered during the pipeline design. The ability of models to continually learn and improve their performance by incorporating additional training data flow, without the need for complete retraining, should be systematically investigated[27,28]. This approach can enhance the cost-effectiveness of AI models and reduce validation time, as demonstrated in Fig. 1c.

One of the greatest challenges in the field of AI digital pathology is achieving generalization. Addressing this issue necessitates a comprehensive understanding of the biological differences and potential variations underlying these challenges. The development and progression of cancer is a process that unfolds in four dimensions (4D), as illustrated in Fig. 1d. This implies that cancer develops within three-dimensional (3D) space In Vivo over time. The process of In Vitro Diagnostics (IVD), i.e., biopsy or surgery, involves extracting a piece of 3D tissue sample from this four-dimensional continuum at a specific time to evaluate the patient's condition. Subsequently, the tissue undergoes various preparation steps, including fixation, sectioning, and staining. In the conventional pathology workflow, these prepared samples are examined and interpreted by highly trained professionals, i.e., pathologists. It is a well-established and validated process that pathologists possess the expertise to discern the biological and pathological difference in tissues, enabling them to provide precise diagnoses for patients.

In the realm of AI digital pathology, the performance of artificial intelligence (AI) models is considerably influenced by the variability inherent in the stages of sample preparation described earlier, as well as by variation during the data acquisition or whole slide imaging phase, as depicted in Fig. 1d. The substantial real-world variations dependent on the sites and the machines lead to a decline in model performance when faced with previously unseen data. It poses a critical challenge in the field, as it hinders models from adapting to and maintaining consistent performance across diverse datasets, clinical centers, and different scanners. It is a principal reason behind the limited adoption and scalability of AI-based diagnostic models in pathology.

Addressing these limitations and designing comprehensive AI diagnostic model development framework that encompass diverse datasets[29], efficient annotation schemes, and consideration of model updateability are crucial steps towards advancing AI-assisted pathology diagnostic models and accelerating their adoption in the healthcare systems. Being motivated by increasing the scalability of AI-assisted diagnosis solutions in clinical practice and boosting the efficiency and reduce pathologists' workloads, we designed a comprehensive pipeline for development of AI pathology diagnostic model, as shown in in Fig. 1a-c. Overcoming the challenge of obtaining high-quality annotated data is essential for advancing AI diagnostic models. The establishment of a high-quality, structurally annotated

database is vital for AI pathology model training, necessitating rigorous, automated quality control (QC) to guarantee data integrity for model development and validation. We developed A!MagQC software quantitatively assesses digital pathology image quality, optimizing the annotation process and enhancing AI diagnostic accuracy, as shown in Fig. 1b. Our solution, A!HistoClouds, is a cloud-based annotation platform that streamlines pathologist annotations and AI predictions, bolstered by a Pathologist-AI Interaction (PAI)[30,31] for semi-automatic annotation and continuous model improvement, as depicted in Fig. 1c. Besides, to address the generalization challenge, we introduced a concept called image appearance migration to handle images from different scanners, as shown in Fig. 1e-f.

We used our study of AI-assisted Gleason Grading on PCa pathological images as an example to demonstrate our AI model development framework. Notably, optimization and generalization are two crucial considerations during the AI diagnostic model development. Our prostate AI diagnostic and grading model was first developed and validated on a dataset scanned by Akoya Sciences and subsequently extended to images scanned by multiple scanners through a set of generalization techniques. Additional clinical validation was conducted with pathologists from Singapore and China to evaluate the proposed method in a real-world application.

## Methods
### Sample preparation
This study obtained prostatectomy and biopsy formalin-fixed paraffin-embedded (FFPE) tissue specimens from the Department of Pathology at the National University Hospital in Singapore, with approval from the National Healthcare Group Domain Specific Review Board (DSRB ID: 2018/01186). The Institutional Review Board waived the requirement for verbal or written informed consent as all specimens used in the study were de-identified. The specimens were processed in accordance with the standard operating procedures of a CAP-accredited histopathology laboratory. H&E-stained sections of 4μm were prepared from tissue blocks of radical prostatectomy and biopsy specimens. The study included a total of 187 prostatectomy specimens (total tissue area 112,400 mm$^2$) and 156 biopsy specimens (total tissue area 7723 mm$^2$) from 214 patients. The patient profiles are listed in Supplementary Table 1.

### WSI scanning and image quality control using A!MagQC
Images were initially acquired using Vectra ® Polaris™ from Akoya Biosciences with bright-field imaging at 0.5μm × 0.5μm per pixel resolution. Since annotation quality and subsequent model development are ultimately dependent on the quality of scanned WSIs, it is critical to have a robust QC system and standard to quantify the variation of sample preparation and tissue types. Variations in color, brightness, and contrast can also occur among different scanners and brands used by different pathology laboratories. To address these issues, we developed an image QC software named A!MagQC to quantitatively assess common image quality issues. A!MagQC is an automated histology image quality assessment tool to identify five common categories of WSI quality issues: out of focus, low contrast, saturation, artifacts, and texture uniformity[32–37]. A image patch is classified as "low quality" if A!MagQC identified two or more quality issues in it. Details of A!MagQC are described in Supplementary Methods and Supplementary Fig. 1.

To study the characteristics of images scanned by different scanners and their impact on AI model performance, 38 prostatectomy specimens were selected and scanned using 5 other scanners (Olympus, KFBio, Zeiss Leica, Philips) at same resolution. Our database, including the WSI from different scanners, was consolidated as Automated Gleason Grading Challenge (AGGC) competition hosted by MICCA2022 and it was also part of the benchmark in a recent paper[29].

### Structured image data annotation using A!HistoClouds
With A!MagQC, we ensured WSIs for annotation are of high quality. A user-friendly, easily accessible, and efficient annotation solution is desired

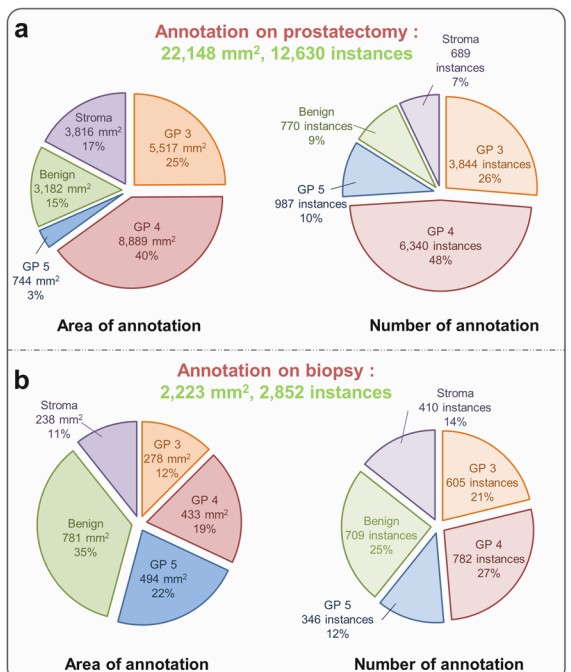

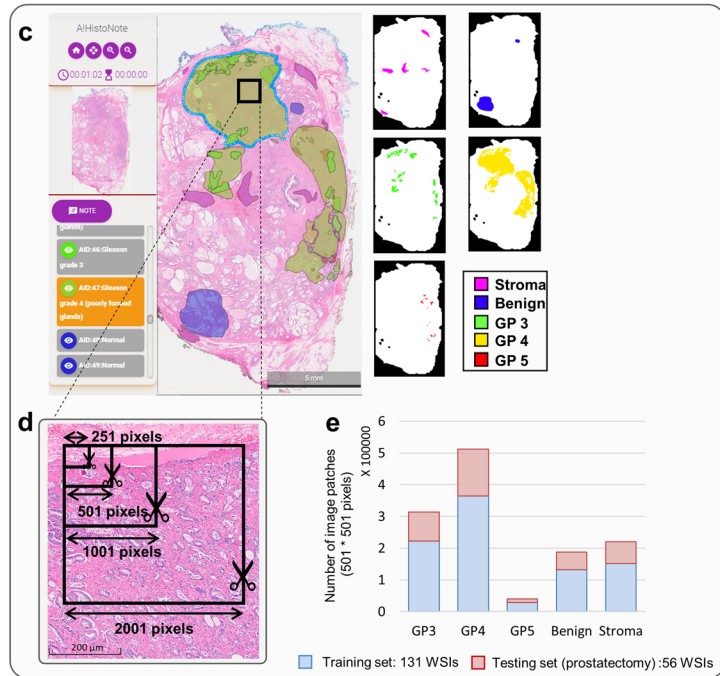

**Fig. 2 | Details of annotated data for training and testing in this study.** The amount of manual annotation performed on prostatectomy and biopsy samples by pathologists is summarized in (**a**) and (**b**), respectively. **c** The annotations of different classes were extracted from A!HistoClouds. **d** To study and optimize patch scale factors, image patches were cropped into different sizes of overlapping patches using a sliding window approach, with a side length of $x$ pixels and a stride of (x-1)/2 pixels, where $x$ = 251 (~125 μm), 501 (~250 μm), 1001 (~500 μm), and 2001 (~1000 μm). The overlap ratio between adjacent image patches is $(x + 1)/2x$. To ensure validity, image patches were extracted only when more than 70% of the area was annotated. **e** Prostatectomy specimens were split into training and testing sets, while biopsy specimens were used for testing only. The same configuration of train-test splitting was applied to datasets of different scales. A bar plot shows the number of image patches of size 501 × 501 pixels in the training and testing set of prostatectomy specimens. Rearranged the lettering so that they precede each description.

for detailed structure annotations. We thus developed a cloud-based annotation platform, known as A!HistoClouds. It provides an image management system (IMS) with different annotation styles, including flexible region of interest (ROI) creation within the image viewer. Details of A!HistoClouds can be found in Supplementary Methods and Supplementary Fig. 2.

Three pathologists from NUH manually annotated the WSIs (each WSI was annotated by one pathologist) using A!HistoClouds to yield total annotation areas of 22,148 mm² (12,630 instances) on 187 prostatectomy specimens and 2223 mm² (2852 instances) on 156 biopsy specimens, as illustrated in Fig. 2a-b. The annotations of different classes were extracted from A!HistoClouds and organized by labels, as illustrated in Fig. 2c. Annotation labels include Gleason pattern 3 (GP3), Gleason pattern 4 (GP4), Gleason pattern 5 (GP5), benign, and stroma tissue. For binary classification, we grouped the GP3, GP4, and GP5 tissue as "Malignant," whereas benign and stroma tissue comprise the "Non-malignant" group.

We used prostatectomy specimens, which provide more abundant information due to their larger size, for training and testing our AI models. Prostatectomy WSIs were split into training and testing sets. Annotated biopsy images were used for testing. After train-test splitting of prostatectomy specimens (described in Supplementary Note 1), an additional evaluation was conducted involving 9 pathologists (5 junior and 4 junior) from 5 hospitals in China. Specifically, 5 junior pathologists individually adjusted the NUH's annotations of each WSI, and the senior pathologists from the same centers further reviewed and made individual adjustments to the annotations made by the junior pathologists. This rigorous process was implemented to ensure the high quality of the testing data, taking into consideration the presence of inter-observer variations in Gleason grading. Finally, the annotations agreed upon by the senior pathologists were used as the "gold standard" to evaluate the model performance.

## AI model development and optimization

Pathologists make diagnoses by examining specimens under a microscope at different magnifications, suggesting that the AI model might also benefit from an optimal magnification for patch classification. To optimize patch scale factors, we trained AI models using patches of different resolution. Each annotated region was cropped into different sizes of overlapping patches using a sliding window approach, as illustrated in Fig. 2d. The patches were then resized to 224 × 224 pixels to fit the input layer, resulting in a resolution of 0.56 (extra high resolution), 1.12 (high resolution), 2.24 (medium resolution), and 4.47 (low resolution) μm/pixel for patches whose original side lengths were 251, 501, 1001, and 2001 pixels, respectively. The number of patches (501*501 pixels) in the training and testing datasets for each class is given in Fig. 2e. The image patches for the training were then organized by type, and representative patches of different classes are shown in Fig. 3a. Data augmentation such as rotation and flipping were applied to each image patch as shown in Fig. 3b. The classification layer in the network was replaced with a Weighted Classification Layer as a class rebalancing strategy, shown in Fig. 3c, where the weights are inversely proportional to the number of image patches to mitigate imbalance in the dataset. To optimize our model's performance, we selected three of the most widely used network architectures in the field for model selection, namely ResNet50, VGG16, and NasNet Mobile. The details of hardware and software used in this work are provided in Supplementary Methods.

During testing, the trained models were applied to every sliding window on test images within the tissue region. The testing process produced a list of scores indicating the predicted probabilities of class labels for each patch, as shown in Fig. 3d, e. We developed a voting policy that considers neighboring patches to determine the predicted label of the overlapping region. The final decision is made based on either the class with the highest votes or the highest average scores. Details of the voting policy can be found in Supplementary Note 2. The final output was compared to the ground truth annotation for performance evaluation, as illustrated in Fig. 3f, g.

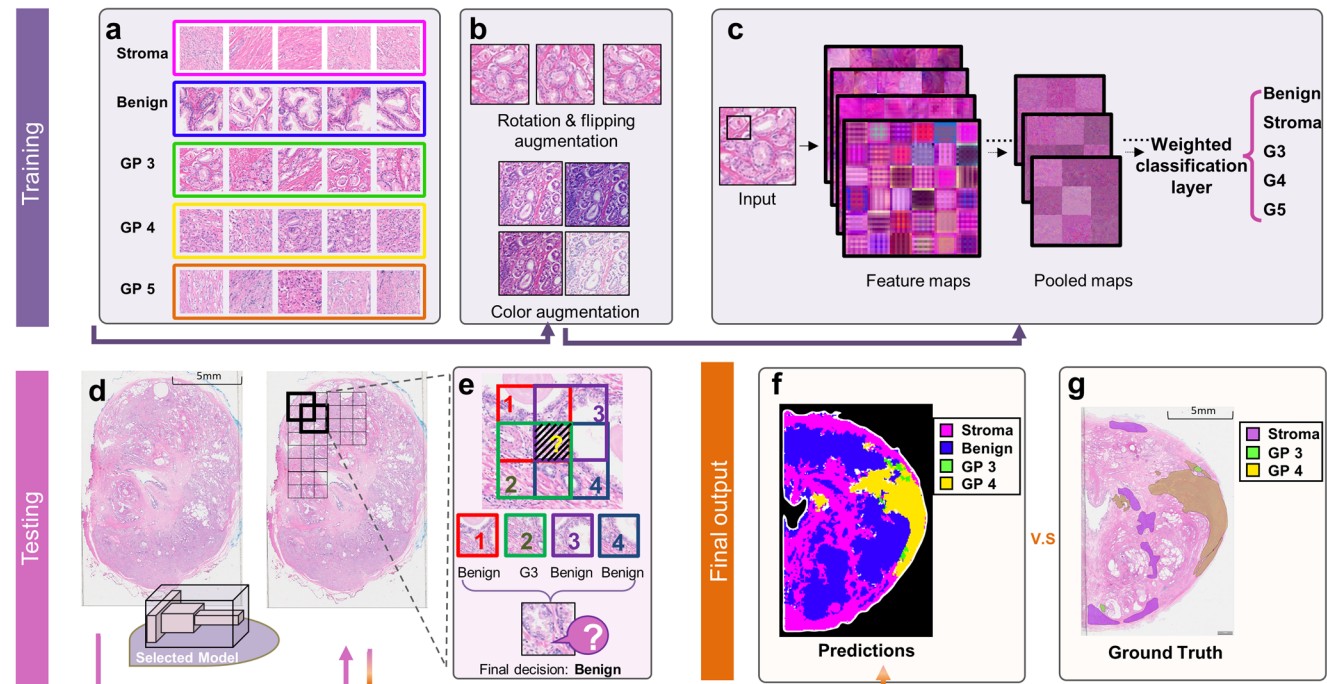

**Fig. 3 | Color augmentation and patch sampling of training and testing of the AI model development.** To prepare the image patches for training, data augmentation was applied to each patch, as demonstrated in (**a**) and (**b**). Besides random rotation and flipping, color augmentation changed the intensity of pixel values randomly within a certain range (details provided in Supp Table 4). The structures of ResNet50, VGG16, and NasNet Mobile were used, and their regular classification layers were replaced with a weighted classification layer. **c** Common structures of the AI model. **d** In the testing process, the trained model was applied to the test image using a sliding window operation with (**e**) a voting strategy. **f** An example of prediction results and (**g**) its corresponding ground truth.

## Image appearance migration and model generalization across different scanners

To improve the generalization of AI models and ensure their adoption and scalability in medical applications, it is essential to understand, quantify and compensate the procedural variations that impact image appearances from sample preparation to data acquisition, while preserving the inherent biological and pathological distinctions. With this understanding, we introduce an approach to tackle these challenges. To validate this concept, we used various scanners to capture images from the same set of glass slides, resulting in images that exhibit significant differences in appearance as in Figs. 1d and 6a. This variation is illustrated by distinct clusters in the RGB color space, as shown in Fig. 1e. By applying techniques such as histogram or Probability Density Function (PDF) matching, we were able to "migrate" the images from different scanners into a unified reference space as shown by the arrows in Fig. 1f. The aim of this approach is to minimize and compensate the variations attributable to scanning and potentially other factors, thereby demonstrating the effectiveness of our method in standardizing image appearances and enhancing model performance across diverse datasets.

Scanner-induced image variations is one of the key variations causing the poor generalizability of AI models. We address this challenge through image appearance migration. The concept is to reduce image inconsistencies and migrate images appearance from different color/feature space to a standard reference space. This transferred space is very similar to the characteristics of the Akoya scanner (baseline), whose images were used for model training. Specifically, we estimated the reference probability density function (PDF) for each RGB channel using 30 WSIs from the Akoya dataset. Images scanned by other scanners are then transformed to match this reference distribution. We further enhanced the generalizability of the models by enlarging the original training dataset using color augmentation during training, as shown in Fig. 3c. The resulting patches from color augmentation were used to simulate the appearance variations of different scanners. We applied three image augmentation configurations, which is designed for the three scanners of clear appearance difference with our reference scanner, to each original image patch, and the details are provided in Supplementary Table 2. The training configurations are the same for the original model and the model with color augmentation.

The annotations were created by pathologists in the images scanned by Vectra Polaris from Akoya Biosciences scanner, and they cannot be directly applied to images scanned by other scanners since the images were not aligned with each other. To resolve this issue, we performed image registration to transfer the annotations created in Akoya-scanned images to the images obtained from other scanners. After registration, the images obtained from other scanners had corresponding ground truth annotations and could be compared with the prediction results to evaluate the performance before and after applying generalization techniques. We achieved accurate image registration, as indicated by the average Structural Similarity Index Measure (SSIM) of 0.99 between the original annotations and registered annotations.

## Pathologist-AI interaction via A!HistoClouds to expedite AI model development: semi-automatic annotation and incremental learning

Initially, the AI model was trained and optimized based on the manual annotations of three pathologists from NUH. This model was known as the base model and serves as a foundation for future development. While more data will be collected to expand the annotation database and improve model performance, manual annotation and retraining the model from scratch is time-consuming and inefficient.

To facilitate further development based on the existing model, Pathologist-AI interaction (PAI) was implemented in A!HistoClouds, as shown in Fig. 1c. During PAI, pathologists perform semi-automatic annotation by correcting pseudo annotations generated by the AI model. Subsequently, these annotations are directly applied to updating the existing model through further training. Outputs of the new AI model serve as a

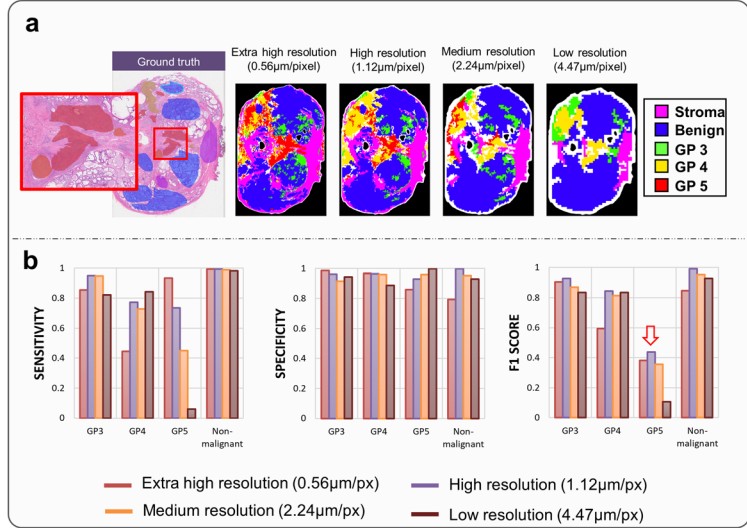

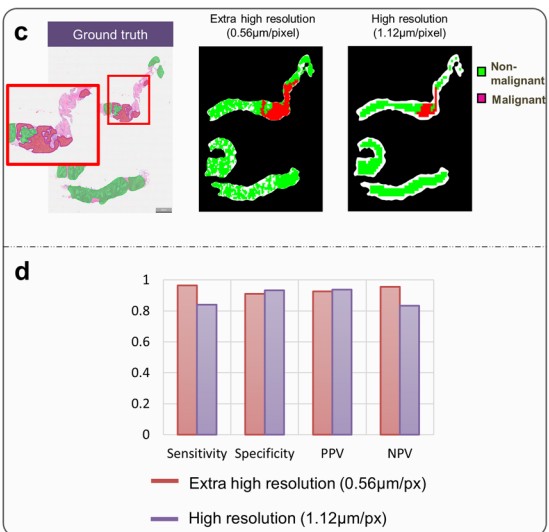

**Fig. 4 | Scale optimization and model performance on prostatectomy and biopsy specimens. a** Model performances at different resolutions and the ground truth annotations. **b** Sensitivity, specificity, and F1 scores. The high-resolution model was selected for processing prostatectomy specimens because its F1 score is the highest for all labels. **c** Similarly, models at different resolutions were applied to biopsy images. **d** Considering the shape and size of the biopsy, a model at an extra-high resolution is more desirable. In particular, part of the biopsy was barely processed at lower resolution but was correctly identified as benign tissue at an extra-high resolution.

starting point of semi-automatic annotation for pathologists in next round of PAI, allowing for a closed loop between the AI model and pathologists.

In this experiment, the base model was applied to 39 slides randomly selected from the testing set to generate pseudo annotations, while the rest of the 17 slides remained as testing images. Pathologists from China and Singapore used A!HistoClouds to correct the pseudo annotations and record the time spent on each image. The corrected annotations were used to update the base model using an incremental learning approach, meaning that the model learned from the new data directly from the base model. Specifically, we froze the top layer of the model and only updated the weights of the bottom layer during training. After training, the base model and updated model were applied to the remaining 17 WSIs in the test set to compare their performance in distinguishing between malignant and benign tissues annotated by NUH pathologists.

### Evaluation metrics

We evaluated the performance on both the annotation-level and the WSI-level. On annotation-level, sensitivity and F1 scores of each class label were calculated for multi-class classification on prostatectomy specimens. For the performance assessment across different scanner dataset, macro average F1 score is computed using the arithmetic mean of F1 scores of different classes. For core needle biopsy samples, we simplified the comparison to binary classification according to clinical needs. Sensitivity, specificity, positive predictive value, and NPV) were used. Pathologists' annotations are non-exhaustive, and each image contains multiple labeled instances, whereas the AI model predicted the label at the pixel level for the entire tissue region. Thus for any instance annotated by pathologists, the corresponding regions in AI output might contain various labels. To reconcile these labels, we adopted a majority voting strategy to select the predicted label that covers the largest pixel area as the final label for that instance in AI output.

On WSI-level of evaluation, we compared the Gleason Grade Group (GG)[38] determined by the AI model and pathologists. GG 1: Gleason Score 6, GG 2: 7(3 + 4), GG 3: 7(4 + 3), GG 4: Gleason Score 8, GG 5: Gleason Score 9 and 10. Quadratic weighted Kappa was used to measure the consistency of two observers.

### Reporting summary

Further information on research design is available in the Nature Portfolio Reporting Summary linked to this article.

## Results

### Scale optimization and evaluation of AI model

After selecting the structures of the AI model (details described in Supplementary Methods and Supplementary Fig. 3), the model performance is evaluated at different resolutions/scales using prostatectomy specimens. A representative image and the results at different scales are shown in Fig. 4a. Four models at different scales are trained using Resnet 50 structures and tested, revealing that scale factors have less impact on the detection of G3, G4, and non-malignant classes. However, the F1 score of G5 increases more than 4-fold with high resolution, as shown in Fig. 4b. Overall, the model trained on high-resolution patches (1.12 μm/pixel) is found to be most effective. It demonstrate high F1 score in predicting GP3 (0.93), GP4 (0.84), GP5 (0.44), and non-malignant tissue (0.99). Of note, the precision, sensitivity and F1 score of predicting non-malignant tissue are all 0.99, which means the model rarely misclassify tumor as benign tissue. Evaluation of the AI model using multiple pathologists' annotations individually is described in Supplementary Methods and Supplementary Fig. 4.

The AI model is also used for pre-screening biopsy samples, and the optimized model is applied to test tumor detection performance. Extra high-resolution images are found to be more suitable for biopsy specimens as they achieve better performance, particularly for sensitivity and negative predictive value (NPV), as shown in Fig. 4c, d. The NPV is 0.96 for 156 biopsy samples, meaning that tumors are very unlikely to exist when the AI model classify samples as non-malignant.

### Image quality assessment using A!MagQC and image appearance migration

Images of the same set of glass slides show remarkable variations when acquired from different scanners, as demonstrated in Fig. 5a. Using the A! MagQC software, we quantitatively assess the quality properties and profiles of images from these scanners, and the low-quality percentage identified by A!MagQC of each image was summarized in Fig. 5b. Our analysis reveals that there are serious quality issues with the image acquired from Scanner C (in order to avoid potential market conflicts, we anonymize the scanner brand and models). After manually reviewing and communicating with the technicians, we confirm the "out-of-focus" issue indicated by the A!MagQC software, which is due to faulty compartment and maintenance issues of Scanner C. Although images acquired from Scanner A and B had more artifacts, the general quality of all images scanned by different scanners are satisfactory and pass the QC test. Specifically, some WSIs have higher "out-

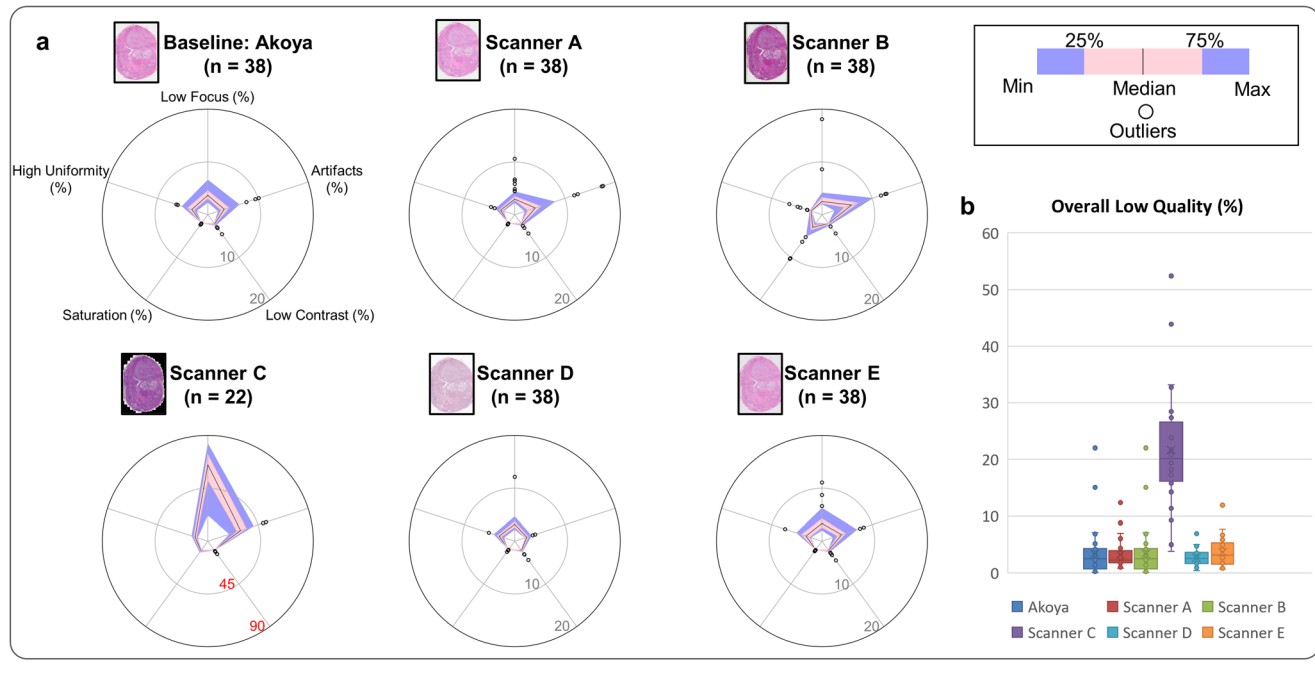

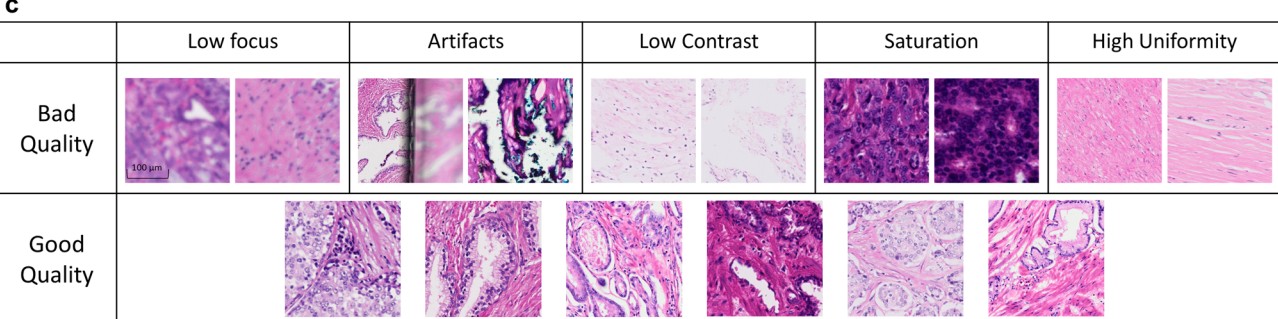

**Fig. 5 | Image Quality Assessment Using A!MagQC. a** A!MagQC assessed the images from different scanners and identified five common quality issues. **b** The percentage of image tiles classified as "Low Quality" in different scanner datasetss, where a tile was considered "Low Quality" if it exhibited more than two quality issues. Notably, images obtained from Scanner C displayed lower quality due to issues related to the scanner's faulty hardware. To validate the A!MagQC results, manual examination was conducted on randomly-selected tiles. **c** provides examples of patches with and without severe quality issues as identified by A!MagQC.

of-focus" scores than others. We perform manual review for all WSIs of comparatively lower quality and confirm that the QC results are accurate. Some example of patches that pass or fail the A!MagQC assessment are demonstrated in Fig. 5c.

Variations in image appearance caused by scanning might impact the performance of the AI model, as shown in Fig. 6a, b. Image appearance migration, specifically employing PDF matching to align images scanned by different scanners with the reference distribution of the Akoya scanner (baseline), is used to reduce scanner-induced discrepancies. The resultant images are demonstrated in Fig. 6c. We assess its effectiveness by measuring the histogram intersection of tissue regions between baseline and image datasets before/after appearance migration, as shown in Supplementary Fig. 5. The results show that image appearance migration greatly increase the similarity between baseline and acquired from other scanners, with almost perfect overlap in histogram intersection for all channels.

**Model generalization for invariant performance across multiple scanners**

Our study shows that unadjusted variations in the intensity and color of commonly used scanners could lead to inconsistent predictions. Notable variations in the imaging of the same glass slides yielded by different scanners, resulting in discrepancies in AI model outcomes despite identical underlying histopathological content. Specifically, Scanner B and C tend to over-diagnose due to darker images, while Scanner D tends to under-diagnose due to lighter images, compared to the baseline.

To improve the generalizability of the AI model across different scanners, we apply image appearance migration to the WSIs and color augmentation to the model training. After these techniques are applied, the AI results across various scanners become more consistent and aligned with the ground truth annotations, as shown in Fig. 6e, f. A marked increase in sensitivity, specificity, and F1-score of tumor grading is observed after applying image appearance migration and color augmentation, as shown in Supplementary Fig. 6a–c. The improvement is particularly pronounced for images scanned by Scanner B, C, and D, whose differences from the baseline are more substantial. However, if images acquired from other scanners are close to the baseline dataset, generalization techniques might not be necessary. In addition, we conduct a comparison of the macro average F1 score when applying image normalization only versus color augmentation only, to evaluate their effectiveness individually as shown in Supplementary Fig. 6d. Our results show that the differences in the macro average F1 score between applying image normalization only, color augmentation only, and both image normalization and color augmentation are comparatively subtle. These findings suggest that both image normalization and color augmentation optimize the generalization performance across images scanned by different scanners.

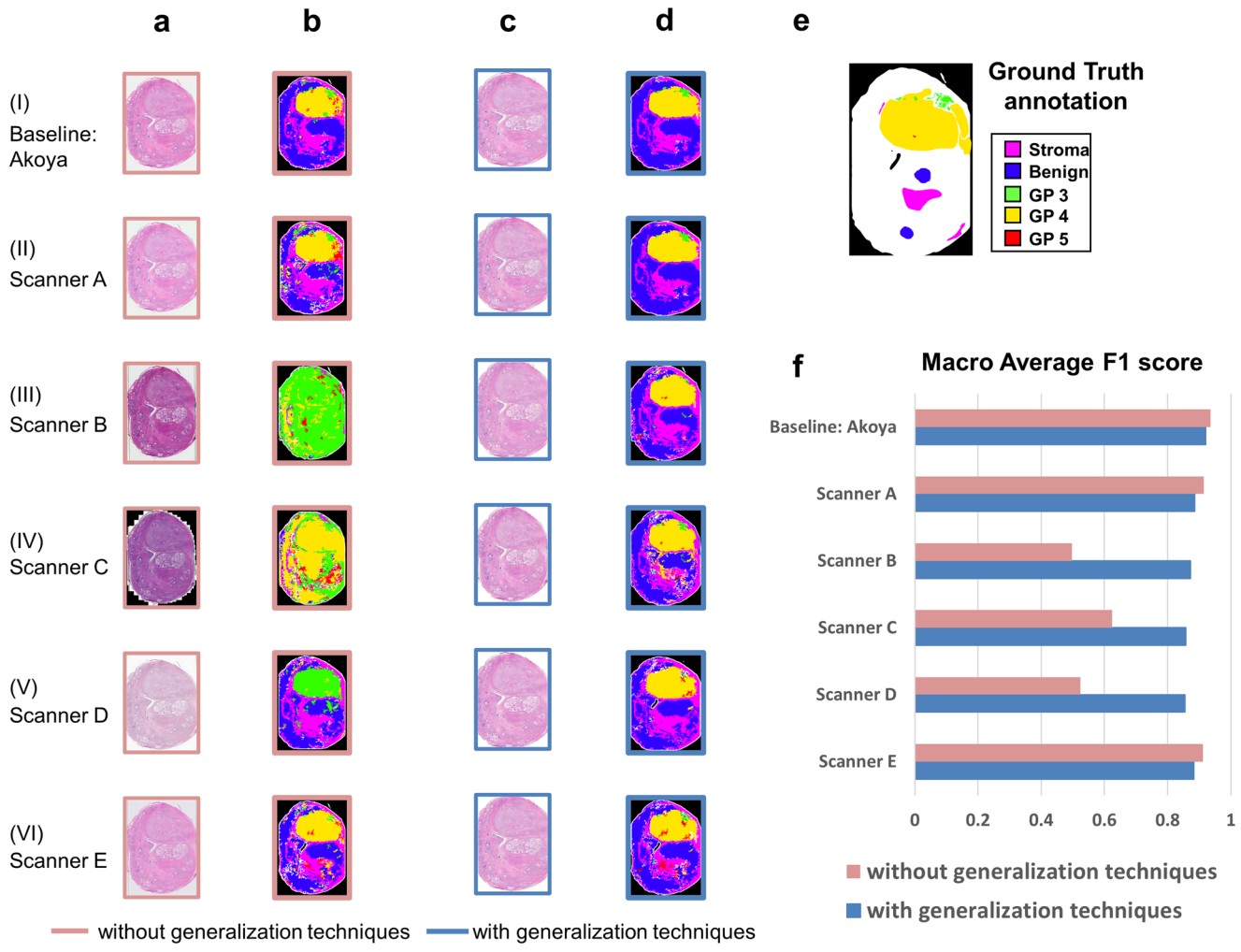

**Fig. 6 | Model performance generalization across images from different scanners.**
**a** Images acquired from different scanners exhibit diverse appearances. Without applying generalization techniques, these variations lead to inconsistent AI results, as shown in (**b**). To mitigate the impact of appearance variations on AI model performance, images from various scanners underwent image appearance migration, reducing appearance differences. Additionally, color augmentation was employed during the generalization process, training the model using the same method as previously described. **c** and **d** display images after appearance migration and their corresponding outputs from AI model with color augmentation, respectively. With the implementation of the generalization technique, the AI outputs demonstrated increased consistency. **e** And the corresponding AI outputs better aligned with the ground truth annotations. **f** The macro average F1 score of each scanner dataset before and after the generalization techniques were applied. Notably, Gleason Pattern 5 was excluded due to the limited availability of annotated GP5 regions in cases scanned by multiple scanners.

### Three-phase clinical validation of AI-assisted diagnosis

We aim to assess whether the use of an AI model can improve the accuracy and efficiency of Gleason grading in histopathology departments. A three-phase clinical validation is conducted involving five pathologists from China and Singapore. In the experiment, we compare traditional microscopic examination (Phase 1), WSI examination without AI-assistance (Phase 2), and WSI examination enhanced by AI-assistance (Phase 3), as shown in Fig. 7a. In phases 2 and 3, pathologists use A!HistoClouds to examine the WSIs. In phase 3, pseudo annotations generated by the AI model, along with Gleason scores and tumor percentages, are provided as references for the pathologists. Pseudo annotations are simplified for easier interpretation and quicker diagnosis based on pathologists' feedback prior to the experiment, as shown in Fig. 7b. Details of this experiment are described in Supplementary Methods.

The accuracy of GG classification is assessed by calculating the Quadratic Weighted Kappa of pathologists A–E across the three phases, using the GGs annotated as references by another four senior pathologists, as indicated in Fig. 7c. In Phase 3 the average agreement level between the AI-assisted diagnosis and Senior Pathologists 2 and 3 is found to be the highest, indicating that the AI-assistance improved the accuracy of Gleason grading.

However, if we consider Senior Pathologists 1 and 4 as the reference, Phase 3 do not consistently achieve the best results, although the difference was only subtle. It is worth mentioning that the agreement level of the AI model and senior pathologists is consistent with that of pathologists A–E across different phases. Almost no difference is observed between Phase 1 and 2 among pathologists A, B, and C in terms of time taken for histological examination, as shown in Fig. 7d. However, there is notable improvement for all 5 pathologists in Phase 3, especially for pathologists A, D, and E, whose average examination time is reduced by 41–58%. With the introduction of AI-assistance in the final phase, the average examination time of pathologists is reduced by 43%, decreasing from 148s (Phase 1) and 147s (Phase 2) to 84s (Phase 3). The p value of the two-sided paired t-test is calculated, indicating a clear decrease in diagnosis time for Pathologists A, D, and E. These results show that using WSI examination with AI-assistance considerably improved the efficiency of Gleason grading without compromising accuracy, suggesting a role as assistant to support pathologists in their diagnostic work.

### Evaluation of pathologist–AI interaction efficacy

The PAI strategy, comprising two components: semi-automatic and incremental learning, is designed to reduce the annotation time of

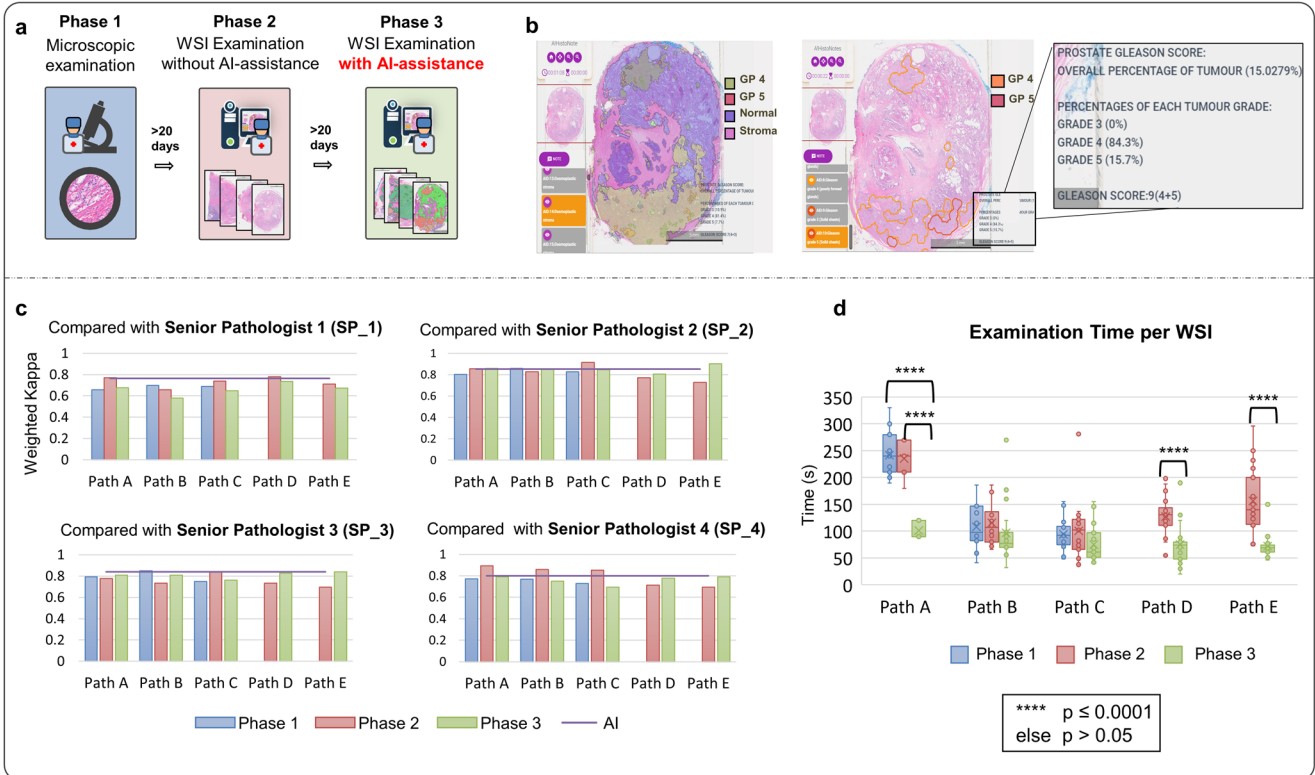

**Fig. 7 | Three-phase clinical validation. a** We designed a 3-phase clinical validation to assess AI-assisted diagnosis by comparing microscopic examination and WSI examination with and without AI-assistance. **b** In Phase 3, the pseudo annotations we presented to pathologists were simplified according to pathologists' feedback, since the original prediction generated by the model contains excess detail). **c** We assessed the accuracy of Gleason Grade Group at different phases by comparing pathologists A–E's readings with those of four senior pathologists. **d** The average time pathologists are required to examine a WSI at different phases, demonstrating that the highest diagnosis efficiency was achieved in Phase 3.

pathologists and upgrade the existing AI model when new data becomes available. We evaluate PAI's effectiveness from two perspectives: the efficiency of semi-automatic annotation and the improvement of AI model performance. First, we record the time required for pathologists to perform annotation using fully-manual and semi-auto methods (see Fig. 8a). We find that semi-automatic annotation was about 2.5 times faster than fully manual annotation. The average annotation speed decreases from 1267s/per image for fully manual annotation to 508 s/per image for semi-automatic annotations and the *p* value of two-sided paired *t* test is 0.0009, indicating substantial decrease as demonstrated in Fig. 8b. From this perspective, semi-automatic annotation supported by our AI model greatly improved a pathologist's efficiency to iteratively integrate new data into the model. We also compare the total annotated areas before and after pathologists' corrections to quantify the quality of pseudo annotations generated by the AI model. As shown in Fig. 8c, most of the pseudo annotations of GP3, GP4, normal, and stroma are preserved, while many GP5 annotations are removed. These findings align with the image dataset's validation results, indicating the AI model's satisfactory performance in identifying GP3, GP4, and non-malignant samples, but relatively poor performance in identifying GP5 due to limited data availability.

More importantly, after feeding the corrected annotations directly into the existing base model for further training (see Fig. 8d), the overall performance of the AI model increases across all metrics, as shown in Fig. 8e, demonstrating its capability of incremental learning without the need for complete retraining. As the AI model's performance will improve continuously through learning from pathologists' corrections, we anticipate that pathologists will require less time and effort to perform semi-automatic annotations, leading to increased accuracy of pseudo-annotations.

## Discussion

We address key unmet challenges in AI-assisted prostate Gleason grading, including the lack of automated image quality assessment, inefficient annotation and model updating, and poor model generalizability. Our solution is a comprehensive pipeline for developing AI models. This workflow comprises several crucial components: (1). A!MagQC performs automated quality control on digital pathology images, ensuring consistent data quality for reliable AI model development. (2) A!HistoClouds facilitates efficient annotation and visualization of AI results, enabling pathologist contributions to model training and benefiting from AI-assisted diagnosis. (3) Pathologist-AI Interaction (PAI) bridges the gap between pathologists and the AI model through semi-automatic annotation and incremental learning, leading to continuous improvement and adaptation. Notably, we also implemented robust generalization techniques, including image appearance migration, to ensure consistent performance across multiple scanner models, which was rarely considered in previous studies. This comprehensive approach, which is generalizable to other AI pathology diagnostic model development, tackles critical bottlenecks in AI-assisted Gleason grading, paving the way for its widespread adoption and transformative impact in clinical practice.

Scanning plays a crucial role in Digital Pathology, and our A!MagQC provides a tool for identifying quality issues in WSIs. The study shows that A!MagQC can also detect the working condition of scanners. It is important to note that the performance of the AI model can be affected by variations in the appearance of images acquired from different scanners. The generalizability of AI models across image scanners is a crucial consideration when deploying the AI model for diagnosis in hospitals using existing pathology workflows and scanners, ideally without the need to retrain the model. It is important to highlight that, while image appearance migration and color augmentation both offer similar generalizability as in Supplementary

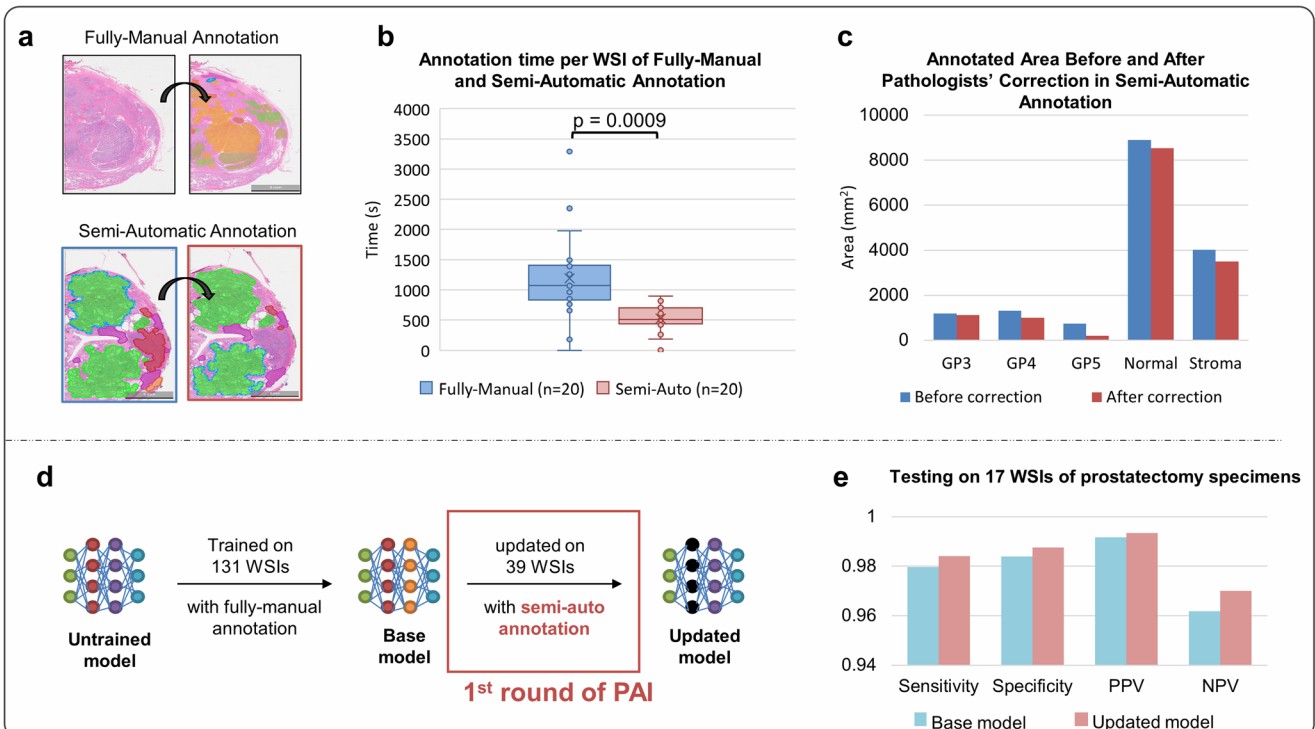

**Fig. 8 | Assessment of PAI.** In this study, the Pathologist-AI interaction (PAI) strategy is evaluated, comprising two components: semi-automatic annotations to enhance pathologists' annotation efficiency, and incremental learning for updating the existing AI model without full retraining. **a** illustrates the comparison between fully-manual annotation where pathologists annotate from scratch, and semi-automatic annotation where pathologists adjust pseudo annotations generated by the AI model. **b** presents a time comparison between fully-manual and semi-automatic annotation methods, while (**c**) compares the annotation area for each class before and after pathologists' corrections. **d** demonstrates the workflow of incremental learning, where the base model is updated using semi-automatic annotations, leading to improved model performance after the 1st round of PAI, as shown in (**e**).

Fig. 6d, image appearance migration may present a more viable option for real-world applications. Firstly, the color augmentation in our study was specifically targeted to mimic the images scanned by scanners B, C and D, rather than being universally. Although models trained with color-augmented data performed well within our study, this approach does not assure generalizability to images scanned by unknown and untested scanners with unforeseen appearances without additional model training. Secondly, color augmentation significantly increases the volume of training data, which leads to increased computational costs, particularly when attempting to simulate images from numerous scanners. In contrast, image appearance migration can be effortlessly applied to images from any source without additional training. This method is regarded as a more straightforward, cost-effective, and robust solution.

By employing the image appearance migration techniques, our approach minimized variations in image presentation. This method enables AI models to precisely identify and assess critical biological, pathological, and cytological features with minimal impact from procedural variations, thereby enhancing their generalizability. Consequently, this approach significantly improves the ability of AI models to generalize across different datasets (including the variable of staining, hospitals and scanning), representing a notable progress in the field of digital pathology. It ensures that AI models can reliably interpret and analyses medical images, regardless of the source or method of preparation, paving the way for more consistent and accurate diagnostic outcomes. Better generalization will potentially improve the adoption and scalability of the AI model applications. We emphasize the need for developing and validating the AI models based on the dataset with real-world variations, as this will allow the model to be implemented in different hospitals and labs. Furthermore, a generalizable model that is tolerant of changes in imaging parameters is crucial for clinical practices, where suboptimal tissue processing and scanning conditions inevitably occur. Our findings strongly suggest that image appearance

migration solution are effective for enhancing the consistency and performance of AI models across different types of scanners.

Although we only investigate the variations caused by scanning in the study, the proposed methods should also be capable to deal with any other variations in image appearance caused by other upstreaming process, e.g., tissue cutting, staining, and sample storage. Image appearance migration should be one of the primary solutions to improve the generalization ability of AI models in DP. Similar concept is able to be generalized to cytology samples, while the solution described in this work may not be directly applicable to cytology images which require more elaborative techniques.

Our AI model shows high accuracy in detecting and grading PCa in both biopsy and prostatectomy specimens, achieving a specificity of over 90% for the more commonly observed GP3 and GP4 samples. Moreover, it can serves as a valuable pre-screening tool, enabling pathologists to quickly identify benign cases and focus on suspicious specimens with greater confidence. This binary model can identify malignancies from a large number of otherwise benign cases, decreasing the workload of pathologists who can then focus on potentially malignant cases. Nonetheless, our model's accuracy in identifying GP5 is relatively low due to its rarity in our dataset, which impede the model's ability to differentiate it from other patterns. Furthermore, we find that GP5 was barely detected at lower resolutions. This is consistent with the fact that pathologists require higher magnification to determine Gleason grading accurately. We expect that gathering more GP5 data in the future will enhance the model's performance for this grade.

The three-phase clinical validation with five pathologists confirms the AI model's effectiveness in improving both the accuracy and efficiency of Gleason grading. Compared to traditional methods, all five pathologists achieve faster examination times with the AI-assistance, without compromising accuracy. Despite being new to WSI examination with AI-assistance in contrast to their familiarity with examination by traditional microscope,

the pathologists benefit substantially from the information provided by our AI model integrated with A!HistoClouds.

The PAI experiment results demonstrate that AI-assistance increases the speed of annotations by 2.5 times compared to manual annotation, thereby reducing pathologists' workload. This feature is crucial for cost-effective and time-efficient AI model training, which typically requires large datasets. By facilitating an iterative feedback loop between pathologists and the AI model via A!HistoClouds and using incremental learning, the AI model can be efficiently trained and validated, leading to a larger and more generalizable database.

Effective presentation of AI-generated data is crucial for rapid human processing and informed decision-making. In the three-phase clinical validation study, the results show that the AI-assistance implemented in A!HistoClouds enables pathologists to locate and grade tumors more efficiently, thereby greatly reducing the time required for pathologists' evaluation. While computers are potentially more powerful than the human brain in processing information, AI-generated data can be too enriched for humans to interpret effectively. In our case, AI processes every pixel in segmentation tasks, creating a large amount of information. We find that presenting all this information to human pathologists can potentially confuse them and slow down decision-making. Therefore, we interact with pathologists to better understand their diagnostic habits and designed a concise and optimal way to present the AI-processed information to facilitate rapid diagnosis. Close collaboration with pathologists enable us to provide useful information to assist them in making faster decisions.

Furthermore, the ground truth annotations visualized in A!Histo-Clouds offer a valuable resource for researchers developing AI models. By providing a quick overview of annotated structures, A!HistoClouds enhances the efficiency of AI solution development and deployment, serving both pathologists and researchers in the digital pathology field. Importantly, the proposed workflow extends beyond the development of AI-assisted Gleason grading models. Its generalizability allows for its wide-ranging adoption in the development of diverse AI-powered digital pathology solutions for various diseases not limited to a recent comprehensive work[29].

Our future work will concentrate on developing AI model capable of learning from multiple annotators with diverse experience level as we realized that variations exist on both annotation- and WSI-levels. Additionally, we aim to validate and optimize our model for a more diverse range of clinical scenarios, including different tissue thicknesses, staining types, and scanners. Lastly, our AI model development workflow, including A!MagQC and A!HistoClouds, can be applied to develop and validate AI-based diagnostics for other cancers.

## Data availability

The image data set has been published for the Automated Gleason Grading Challenge (AGGC) 2022 as a registered MICCAI 2022 challenge. The dataset can be downloaded at the challenge website: https://aggc22.grand-challenge.org. The data can be used under an Attribution-NonCommercial-ShareAlike 4.0 International (CC BY-NC-SA 4.0) license. Anyone who uses the data should cite the current article. All source data are available in Supplementary Data.

## Code availability

Access to the code is restricted due copyright considerations and anticipated commercialization prospects. To request access, please send an email to Dr Weimiao Yu: yu_weimiao@bii.a-star.edu.sg/wmyu@imcb.a-star.edu.sg.

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

## Acknowledgements

This project is funded by Exploit Technologies Gap-funded Project: "AI based H&E image analysis for prostate cancer staging" (EPTL/18-Gap029-R20H) and A*STAR BMRC ATR Grant awarded to Dr. Weimiao Yu. It is also jointly supported by Shanghai "Rising Stars of Medical Talents" Youth Development Program (SHWRS2020_087) and Deep Blue 123 Free Exploration Program of Changhai Hospital (2019SLZ001) and Deep Blue medical Program of The Second Military Medical University (2021), awarded to Dr. Jing Zhang. Additionally, it receives funding from Innovation & Enterprise (I&E) Gap-funded Project (I22D1AG067) titled "Annotation and Extraction of Targeted Tumor Regions from Difficult Cancer Tissue Sections Using A! HistoNotes Digital Annotation Platform" awarded to Dr. Ong Kok Haur. Additionally, we would like to express our sincere gratitude to Olympus, KFBio, Zeiss, Leica, and Philips for their valuable assistance in WSI scanning.

## Author contributions

W.M.Y. and S.Y.T. conceived the study, supervised the research together with S.H., contributed to project coordination and administration. S.H., W.M.Y., and H.H. collected the data. B.X.L., B.X.C., Q.W., K.G. contributed to the WSI scanning. X.M.H. contributed to model development and validation, experiment design, data analysis. S.H. and O.K.H. contributed to the development and maintenance of A!HistoClouds and provided informatics support. L.G. contributed to the development of A!MagQC, and together with L.J.L. provided informatics support. S.H., K.W.L., C.L.T. performed annotations and participated in the clinical experiments. X.H.Z., C.C.Z., Y.H.Z., J.Z., J.H., H.F.Z., H.L.G., L.J.Y., X.X.W., X.Y.C., H.T.C., X.Z.Y., Y.B.S., Z.L.H., Y.Y.S. and W.Y.C. contributed to the data annotation and clinical experiments. X.M.H. wrote the manuscript under the supervision of W.M.Y. with the contribution of all other authors. D.M.Y., H.D.L., G.M., X.J., W.Z.S., D.L.C., S.J.S., H.K.L., S.S., W.M.Y., and S.Y.T. contributed to the review and revision of the manuscript.

## Competing interests

The authors declare no competing interests.

## Additional information

¹Computational Digital Pathology Lab, Bioinformatics Institute, A*STAR, Singapore, Singapore. ²Computational & Molecular Pathology Lab, Institute of Molecule and Cell Biology, A*STAR, Singapore, Singapore. ³Department of Pathology, National University Hospital, National University Health System, Singapore, Singapore. ⁴Department of Psychiatry and Behavioral Sciences, UCSF Weill Institute for Neurosciences, University of California, San Francisco, USA. ⁵Department of Pathology, Nanfang Hospital and Basic Medical College, Southern Medical University, Guangzhou, Guangdong Province, China. ⁶Guangdong

Province Key Laboratory of Molecular Tumor Pathology, Guangzhou, Guangdong Province, China. [7]Department of Pathology, The 910 Hospital of PLA, QuanZhou, Fujian Province, China. [8]Institute for AI in Medicine, School of Artificial Intelligence, Nanjing University of Information Science and Technology (NUIST), Nanjing, Jiangsu Province, China. [9]Department of Pathology, Shanghai Changzheng Hospital, Shanghai, China. [10]Department of Pathology, Zhongshan Hospital, Fudan University, Shanghai, China. [11]Department of Pathology, Hebei General Hospital, Shijiazhuang, Hebei Province, China. [12]Department of Pathology, Fudan University Shanghai Cancer Center, Shanghai, China. [13]Department of Pathology, Changhai Hospital of Shanghai, Shanghai, China. [14]Ningbo KonFoong Bioinformation Tech Co. Ltd, Ningbo, Zhejiang Province, China. [15]Vishuo Biomedical Pte Ltd, Singapore, Singapore. [16]Cancer Center, Department of Pathology, Zhejiang Provincial People's Hospital, Affiliated People's Hospital, Hangzhou Medical College, Hangzhou, Zhejiang Province, China. [17]Key Laboratory of Endocrine Gland Diseases of Zhejiang Province, Hangzhou, Zhejiang Province, China. [18]Clinical Research Center for Cancer of Zhejiang Province, Hangzhou, Zhejiang Province, China. [19]Department of Urology, Fudan University Shanghai Cancer Center, Fudan University, Shanghai, China. [20]Shanghai Genitourinary Cancer Institute, Shanghai, China. [21]Department of Oncology, Shanghai Medical College, Fudan University, Shanghai, China. [22]Institute of Developmental and Regenerative Medicine, University of Oxford, Oxford, UK. [23]Department of Pathology, Yong Loo Lin School of Medicine, National University of Singapore, Singapore, Singapore. [24]These authors contributed equally: Xinmi Huo, Kok Haur Ong. [25]These authors jointly supervised this work: Susan Swee-Shan Hue, Weimiao Yu, Soo Yong Tan. ✉e-mail: swee_shan_hue@nuhs.edu.sg; yu_weimiao@bii.a-star.edu.sg; wmyu@imcb.a-star.edu.sg; pattsy@nus.edu.sg

