## [Peer Review File · Communications Medicine]

Reviewers' comments:

Reviewer #1 (Remarks to the Author):

The authors develop and describe a system to perform WSI QC, cloud-based and distributed annotations, and AI model development (including utilization of semi-automatic annotations and ability to update the model with new annotations). The authors also provide information about inter-scanner generalizability of the AI model and perform an AI-assistance reader study.

Notable strengths of this work include the specific attention to image quality (via an image quality component of the pipeline) as well as comparison of multiple factors, from different network architectures to evaluation on different scanners. However, development and validation of the Gleason Grading algorithm itself appears consistent with many works previously reported for the same task; also, performance reporting is done on a "patch-level" which makes it hard to compare to prior works or evaluate clinical relevance - both of which are more often considered at the specimen or case-level. Lastly, there are intriguing elements of building a pipeline that includes an annotation tool, but the validation of these individual components is not necessarily a focus of the results presented. While the reader study is interesting, several prior prostate pathology reader studies have also been described, thus, the more novel validation and analysis would be on the QC pipeline, the scanner variability, and impact of iterative annotations on model performance. Each component of this work likely warrants a deeper analysis to understand validity, utility, and to enable useful conclusions. As such, the paper in its current form does not seem to sufficiently validate or advance our understanding of how this work might be used in clinical research or clinical practice beyond what has already been shown for other published tools and algorithms.

Comments:

1) The QC component is potentially a more interesting and novel contribution than the Gleason Grading algorithm development and evaluation itself. Validation and evidence that the QC pipeline is accurate and useful would be an important addition to improve the impact of this work.

Some examples are described by:

Zarella et al. - <https://www.ncbi.nlm.nih.gov/pmc/articles/PMC9327504/>

Norgan et al. - https://prismic-io.s3.amazonaws.com/pramana/69e0faa1-7de1-4106-8293-73fd499d94e3_pramana-mayo-clinical-case-study.pdf

Kholberger et al. - <https://pubmed.ncbi.nlm.nih.gov/31921487/>

2) The scanner variability evaluation is also a useful component of this work. However, more prominent emphasis and description of these data should be provided as currently there is not enough information to draw any meaningful conclusions about differences across scanners.

3) Related to 1 and 2 above, figure 5 could be modified into separate figures or tables to provide more quantitative and informative data corresponding to the columns B, E, F, and G - which are currently somewhat qualitatively interesting and creative, but not as informative as more quantitative analysis and comparisons might be.

4) There does not appear to be a description or justification of any statistical analyses or

comparisons performed, leaving it hard for the reader to interpret the significance/importance of the results - this includes the QC pipeline performance, inter-scanner comparison, and reader study analysis as examples.

5) The emphasis on the Gleason Grading model and validation itself could be de-prioritized in favor of emphasizing the more novel components of the work - eg. QC pipeline, scanner comparison.

6) An additional aspect that could make this work more novel and informative would be comparison of the annotation process itself to existing annotation workflows - demonstrating other metrics of quality such as resulting model performance would be an important contribution in the field. Currently only the annotation time is provided, but information on whether the iterative annotation process improved model performance is not provided.

7) Given the precedent and benchmarks on this well-studied AI application (prostate pathology) - specimen level performance should also be reported for the model (in addition to the patch-level metrics provided).

8) Minor: The use of the “!” in the tool name is creative and adds an element of fun or excitement, but it is quite distracting visually in reading about the tool - would suggest considering an alternative name for easier viewing.

Reviewer #2 (Remarks to the Author):

In this manuscript the authors describe the creation of a digital pathology workflow to utilize artificial intelligence (AI) to help screen and Gleason grade prostate specimens with prostate cancer. The authors incorporate a quality control software to assess image quality of whole slide images and created a cloud-based annotation software that they integrated into their AI model in a feedback loop. They found that this digital pathology workflow resulted in good sensitivity and specificity for tumor detection, and claim it increases the speed in assigning Gleason scores by pathologists.

1. As the authors note, in the last few years there have been an increasing number of manuscripts focusing on utilizing AI to more efficiently detect and Gleason grade prostate adenocarcinomas. The authors of the current manuscript have built an AI algorithm to do this as well, but incorporated a couple of relatively unique features. One of these is a formal assessment of the quality of the whole slide images (WSI) that are utilized. The authors correctly note that WSI vary in quality and can exhibit a number of artifacts that can impact the accuracy of an AI algorithm. They decided to address this by building A!MagicQC, a quality control software to make sure that only WSI of sufficient quality were selected for this study. My concern with this approach is that while QC assessments in general are a good idea, by using it to only select out certain WSI, this may limit the practical utility of this model. In real life, WSI will occasionally have tissue folds, out-of-focus scans, staining artifacts, etc. For certain specimens (i.e. needle core biopsies) the tissue is limited, and trying to cut deeper into block to get a better slide without folds, etc. may not be feasible, and the tissue must still be assessed for clinical care. If only high quality WSI are selected to build the algorithm, were only high quality WSI utilized to test the algorithm as well? If so, then if the AI algorithm is applied to WSI of all-comers, are the tumor detection statistics as robust?

2. One of the other more unique features of this digital pathology workflow was the creation of the Pathologist-AI Interaction (PAI) utilized as a feedback loop to directly be applied to the model for further training. This is a great idea and can theoretically improve the efficiency of building and maintaining AI algorithms. My one question about this is whether the authors can clarify how the accuracy of these annotations was confirmed? Interobserver variability is a known confounder in prostate cancer grading, particularly in assessment of Gleason pattern 4. Were multiple pathologists assessing the same WSI and the annotations incorporated into feedback loop if the annotations matched? If different pathologists placed different annotations on the same WSI were they adjudicated in any manner? A continuous feedback loop modifying an AI algorithm is great as long as the accuracy of the annotations are maintained.

3. The authors in this study describe several different pathologists manually annotating images independently for Gleason patterns 3, 4, 5, benign, and stroma on prostatectomy specimens. They also note that the “ground truth” utilized was the Gleason score retrieved from the hospital records, which presumably are the original pathology reports. While the original pathology report for each biopsy specimen would primarily be rendered on a slide-level, and should theoretically match the annotated results on each WSI, the reports for the prostatectomies would assign Gleason grades by incorporating proportions of tumor patterns across many different slides. Therefore, the ground truth for the prostatectomy cases as a whole would not necessarily be expected to match the annotations on individual WSI for these cases. The authors acknowledge this in the Supplemental Materials Section 4 (lines 141-142), but did not mention how they addressed this potential discrepancy. Did they only select WSI from prostatectomy cases with Gleason pattern proportions that matched the original diagnosis? Or did pathologists annotate each WSI in the prostatectomy cases to establish a new ground truth for these cases? It is not clear how many WSI were selected from each prostatectomy or biopsy case, but if a single Gleason score was applied as ground truth across every slide in a prostatectomy case that would seem potentially problematic in terms of assessment of accuracy.

4. In prostate biopsies, in addition to benign or outright malignant diagnoses, more indeterminate diagnoses such as “atypical small acinar proliferation”, or “high-grade prostatic intraepithelial neoplasia” are rendered not infrequently, particularly in biopsy cases, and have impact on clinical decision-making. Where were these diagnoses categorized? Were they slotted under a “non-malignant” ground truth or “malignant” ground truth? Or were these cases excluded from this study?

5. The 3-phase clinical validation experiment involves multiple pathologists assessing 1) glass slides, 2) WSI without AI annotations and then 3) WSI with AI annotations regarding Gleason grading etc. In the Supplemental Section 4 it is noted that 19 slides were examined by each pathologist, with a washout period of at least 20 days. Does this mean the 19 slides examined by each pathologist were the same in all three of these phases? If this is the case, was the order in which the pathologist reviewed these three phases always the same, or did they vary? i.e. did some of them review the WSI with AI annotations first, glass slides second, WSI without AI annotations third, etc? If all 19 slides are identical across all phases of this experiment, and all pathologists reviewed the slides in the same order, even with a washout period of 20 days I wonder if some of the decreased time to render Gleason scoring by the third phase of the study may be a result of recall.

Overall this was an interesting study with a couple of unique features added to the digital pathology workflow that may have the potential to improve the efficiency of prostate cancer identification and

Gleason grading. Clarification of some of the above points may be helpful to put some of the current claims in context.

Reviewers' comments:

Reviewer #1 (Remarks to the Author):

The authors develop and describe a system to perform WSI QC, cloud-based and distributed annotations, and AI model development (including utilization of semi-automatic annotations and ability to update the model with new annotations). The authors also provide information about inter-scanner generalizability of the AI model and perform an AI-assistance reader study.

Notable strengths of this work include the specific attention to image quality (via an image quality component of the pipeline) as well as comparison of multiple factors, from different network architectures to evaluation on different scanners. However, development and validation of the Gleason Grading algorithm itself appears consistent with many works previously reported for the same task; also, performance reporting is done on a "patch-level" which makes it hard to compare to prior works or evaluate clinical relevance - both of which are more often considered at the specimen or case-level. Lastly, there are intriguing elements of building a pipeline that includes an annotation tool, but the validation of these individual components is not necessarily a focus of the results presented. While the reader study is interesting, several prior prostate pathology reader studies have also been described, thus, the more novel validation and analysis would be on the QC pipeline, the scanner variability, and impact of iterative annotations on model performance. Each component of this work likely warrants a deeper analysis to understand validity, utility, and to enable useful conclusions. As such, the paper in its current form does not seem to sufficiently validate or advance our understanding of how this work might be used in clinical research or clinical practice beyond what has already been shown for other published tools and algorithms.

Reply: Thank you for your constructive comments. We would like to share the following thoughts with you, and it will be probably helpful to position this work.

As we understand, despite the achievements and developments of AI-based pathology diagnostic solution made in the past decades, the overall adoption of the AI models in healthcare system is still poor. There are many papers published on the prostate diagnosis as well as other cancers in the past 5-10 years. Many of the published academic papers often have the following limitations: 1.) data set is often small; 2.) annotation scheme does not directly solve the diagnostic problem and difficult to support the clinical needs; 3.) collected data does not contain sufficient variations, for example using only 1-2 scanners without generalization solution designed in the pipeline, thus the developed models are often poor in generalization; 4.) The experimental/product design is not comprehensive and general, etc.

The achievements in the past years have very few successful examples of commercialization and medical adoption. Lack of comprehensive and systematic design of the studies on AI digital pathological diagnosis is probably one of the reasons. FDA and other regulation bodies certainly have concerns of providing

market entrancing permission, but a recent encouraging achievement is that PAIGE.AI prostate diagnostic solution developed using Leica and Philips scanners was cleared with FDA in Sept 2021. However, its real-world commercialization is not successful, at least for now. There are still many technical and non-technical challenges, but one of the key reasons is that the current traditional, partially digitalized, pathological ecosystem is not able to well support such clinical adoption. Our work is actually motivated by the above facts in the field and we hope to use our prostate project as a showcase to provide an overview what is a comprehensive pipeline when we want to design a AI pathology diagnostic model which might be easier for future adoption. We hope our work can provide an insight on this aspect, which is reflect by the title of this work.

We appreciated that you highlighted the novelty and depth of the QC component and the analysis on scanner variability. Currently we don't have a standard to specify what kind of image quality is suitable for AI to make an accurate diagnosis. Without such "goalkeeper", it is risky to feed AI model which a WSI of very low quality and expect a reliable diagnosis. Pathology associations and societies are already aware of the importance of WSI image quality for the AI-assistive diagnosis. A number of guidelines are drafted and discussed. Such initiatives are enhanced but still based on previous traditional quality control in classical pathology SOP in a descriptive way. From our computational point view, the QC for AI diagnosis can be automated in a quantitative way. Of course, the criteria between automated quantitative assessment and human eye qualitative evaluation should be close enough, saying generally in practice hospital pathological departments have around ~1-3% of slides are not suitable diagnosis. Our QC solution A!MagQC is integrated into our prostate model development pipeline. It is still coarse, and more efforts are needed, such as detecting more types of sample preparation issues, and building an association between the qualitative/descriptive quality evaluation and automated quantitative assessment. We hope soon we shall have a fully quantitative, automated and reliable industrial QC standard to AI diagnosis solution. Some of our work on the QC is still ongoing and we will publish our data and conclusion in our coming papers.

Certainly, patch level training/learning is not novel as the reviewer mentioned. However, the impact of different patch sizes, *i.e.*, the scale factors, to the classification performance is not really well addressed in our opinion, at least for prostate cancer grading. In our works, we provided the results which shows smaller patch size is preferred for G5 regions. The baseline of making accurate specimen/patient level assessment is to accurately recognize each small patch. While yes, we take into account of reviewers' suggestion and provided the case level results, *i.e.*, the Gleason Score.

We have taken these comments into consideration and made updates to the manuscript accordingly. We have enriched the contents about A!MagQC and scanner variability, providing illustrative examples and meaningful conclusions. (1) We have manually reviewed the WSIs of comparatively lower quality identified by A!MagQC to validate the QC results. (2) We conducted additional experiments and separated the impact of image appearance standardization and color argumentation. We have demonstrated the impact of image appearance standardization for images scanned by 6 different scanners in "**Figure 6 Model performance generalization**

across images from different scanners". (3) We have quantified and investigated different generalization techniques by comparing the AI model performances on images scanned by multiple scanners. We are currently conducting another independent study with our pathologists, many of whom are also the co-authors of this work, to investigate the correlation between the automatic "quantitative" quality metrics and pathologists' "qualitative" assessment of image quality. We will soon publish this study in another independent paper.

In addition, to address the concern regarding evaluation, we have now included "WSI-level" evaluation of model performance. Previously, we had reported performance on an "annotation-level" as Gleason Score information by pathologists for each slide was not available back then. Recently, we have collaborated with 9 pathologists (5 junior, 4 senior) from 5 hospitals to perform Gleason grading on test images of prostatectomy specimens (56 WSIs). This allows us to compare the Gleason Grade Group determined by the AI model and pathologists using Quadratic Weighted Kappa, making our evaluation more clinically relevant and comparable to previous studies.

Again, the constructive comments from Reviewer 1 are highly appreciated. We substantially improved this work according to your suggestions. More details will be provided in the following replies to specific questions.

Comments:

1) The QC component is potentially a more interesting and novel contribution than the Gleason Grading algorithm development and evaluation itself. Validation and evidence that the QC pipeline is accurate and useful would be an important addition to improve the impact of this work.

Some examples are described by:

Zarella et al. - <https://www.ncbi.nlm.nih.gov/pmc/articles/PMC9327504/>

Norgan et al. - https://prismic-io.s3.amazonaws.com/pramana/69e0faa1-7de1-4106-8293-73fd499d94e3_pramana-mayo-clinical-case-study.pdf

Kholberger et al. - <https://pubmed.ncbi.nlm.nih.gov/31921487/>

Reply: Thank you for providing the references. We acknowledge the importance of validating the QC pipeline and have incorporated manual examination to review and confirm the QC results. We have also engaged in discussions with pathologists to understand their requirements, considering that QC standards can be subjective. As mentioned, there are a number of initiatives to push image quality standards by different pathology associations and government agencies, while all the documents are very much descriptive. More studies are needed to convince the pathologists that such qualitative standard can be quantified and automated. This will ease the "goalkeeper" function for the AI database development in the future. Afterall, not all the slides are suitable for human nor AI diagnosis, even the percentage is low. The validation of QC results still necessitates manual review. Please refer to the **section " Image Quality Assessment Using A!MagQC and Image Appearance Standardization" in Results** for further details. We understand that further analysis is required to thoroughly investigate the QC pipeline and we keep working in this direction. Currently, we are in the process of evaluating the QC software by applying

it to WSIs of different diseases collected from multiple hospitals and scanned by various scanners. More comprehensive information will be included in our upcoming paper.

2) The scanner variability evaluation is also a useful component of this work. However, more prominent emphasis and description of these data should be provided as currently there is not enough information to draw any meaningful conclusions about differences across scanners.

Reply: We have conducted a more comprehensive analysis and drawn meaningful conclusions regarding scanner variations and model generalization. The scanning appearance and quality will impact the given optimized model training using other image data from other scanners. However, such performance variation is possible to be compensated when reference data is available. This will potentially improve the generalization of the AI model application. This includes quantifying the variations in image appearances, evaluating the effect of image standardization, and assessing the influence of different generalization techniques. For further details, please refer to the **sections "Image Quality Assessment Using A!MagQC and Image Appearance Standardization" and " Model Generalization for Invariant Performance Across Multiple Scanners" in Results.**

3) Related to 1 and 2 above, figure 5 could be modified into separate figures or tables to provide more quantitative and informative data corresponding to the columns B, E, F, and G - which are currently somewhat qualitatively interesting and creative, but not as informative as more quantitative analysis and comparisons might be.

Reply: Figure 5 in last version has been separated into **"Figure 5 Image Quality Assessment Using A!MagQC"** and **"Figure 6 Model performance generalization across images from different scanners"**. Figures has been improved as suggested. More quantitative and informative results is now included.

4) There does not appear to be a description or justification of any statistical analyses or comparisons performed, leaving it hard for the reader to interpret the significance/importance of the results - this includes the QC pipeline performance, inter-scanner comparison, and reader study analysis as examples.

Reply: We have added the **section "Evaluation Metrics" in Methods** to describe the metrics we used. Previously, this section was included in Supplementary.

5) The emphasis on the Gleason Grading model and validation itself could be de-prioritized in favor of emphasizing the more novel components of the work - eg. QC pipeline, scanner comparison.

Reply: We agreed that the analysis of scanner variations and QC pipeline are novel aspects of our study. We have provided a more comprehensive and meaningful analysis of the manuscript. However, regarding the Gleason Grading model, we have not deprioritized its validation, as it provides persuasive and illustrative evidence for the model's performance

6) An additional aspect that could make this work more novel and informative would be comparison of the annotation process itself to existing annotation workflows - demonstrating other metrics of quality such as resulting model performance would be an important contribution in the field. Currently only the annotation time is provided, but information on whether the iterative annotation process improved model performance is not provided.

Reply: We appreciated the comments. It is not really feasible to compare our annotation with other annotation pipelines. The key reason is the lack of pathology resources, i.e., pathologists. The annotation itself took a substantial amount of time even though our system is pretty well designed. In Singapore, we paid for the time of clinical pathologists. They lack motivation of repeat the similar annotation work in another system. However, we compared the annotation time between fully manual annotation and semi-automatic annotation, and our findings confirm that semi-automatic annotation is significantly more efficient. Additionally, the improvement of the resulting model after incorporating semi-automatic annotations has already been addressed in the previous version's Results section under "Evaluation of Pathologist-AI Interaction Efficacy". We have demonstrated that model performance on "annotation-level" (PPV, NPV, sensitivity, and specificity of tumor detection) improved after integrating data from semi-automatic annotation in **Figure 8 Assessment of PAI**". To further validate the Pathologist-AI interaction, we plan to conduct a larger-scale multi-center study.

7) Given the precedent and benchmarks on this well-studied AI application (prostate pathology) - specimen level performance should also be reported for the model (in addition to the patch-level metrics provided).

Reply: We acknowledge the importance of reporting model performance on "WSI-level" (i.e. Gleason Score or Grade Group), and we have taken steps to address this concern. As mentioned earlier, initially we did not have the Gleason Score information determined by pathologists for each slide, and thus the performance reporting was limited to "annotation-level" only. However, in the past few months, we have invited nine pathologists from five hospitals to independently perform Gleason grading on the test images of prostatectomy specimens, resulting in Gleason Score determination for 56 WSIs. This has allowed us to include "WSI-level" evaluation in our analysis. We have utilized Quadratic Weighted Kappa to compare the Gleason Score calculated based on AI-generated annotations with those determined by pathologists, making our evaluation more clinically relevant and comparable to previous studies. For detailed information, please refer to the **Supplementary Section V: Evaluation of AI model on annotation-level and WSI-level using multiple pathologists' annotations as reference** in the Supplementary.

8) Minor: The use of the "!" in the tool name is creative and adds an element of fun or excitement, but it is quite distracting visually in reading about the tool - would suggest considering an alternative name for easier viewing.

Reply: We took it into consideration, however, to keep the naming of consistent of our work, including other release documentation, webpages, please allow us to keep such small innovativeness on naming. Again, we are grateful for your valuable

comments and suggestions.

Reviewer #2 (Remarks to the Author):

In this manuscript the authors describe the creation of a digital pathology workflow to utilize artificial intelligence (AI) to help screen and Gleason grade prostate specimens with prostate cancer. The authors incorporate a quality control software to assess image quality of whole slide images and created a cloud-based annotation software that they integrated into their AI model in a feedback loop. They found that this digital pathology workflow resulted in good sensitivity and specificity for tumor detection, and claim it increases the speed in assigning Gleason scores by pathologists.

Reply: Thanks for your encouraging comments and constructive suggestions.

1. As the authors note, in the last few years there have been an increasing number of manuscripts focusing on utilizing AI to more efficiently detect and Gleason grade prostate adenocarcinomas. The authors of the current manuscript have built an AI algorithm to do this as well, but incorporated a couple of relatively unique features. One of these is a formal assessment of the quality of the whole slide images (WSI) that are utilized. The authors correctly note that WSI vary in quality and can exhibit a number of artifacts that can impact the accuracy of an AI algorithm. They decided to address this by building A!MagicQC, a quality control software to make sure that only WSI of sufficient quality were selected for this study. My concern with this approach is that while QC assessments in general are a good idea, by using it to only select out certain WSI, this may limit the practical utility of this model. In real life, WSI will occasionally have tissue folds, out-of-focus scans, staining artifacts, etc. For certain specimens (i.e. needle core biopsies) the tissue is limited, and trying to cut deeper into block to get a better slide without folds, etc. may not be feasible, and the tissue must still be assessed for clinical care. If only high quality WSI are selected to build the algorithm, were only high quality WSI utilized to test the algorithm as well? If so, then if the AI algorithm is applied to WSI of all-comers, are the tumor detection statistics as robust?

Reply: Yes, the reviewer certainly highlights an important point about the image quality and the model performance. This is relevant with the generalization of the model in the future application. Here allow us to share some of our considerations on this aspect.

We understand that the image quality will impact the modelling training. The general term for “image quality” has two different aspects: 1. The WSI image contains the defects which will affect the diagnosis in terms of human and AI, such as dust, bubble, folding and broken. In such we call it “image quality” issues and this is the quality control handled by A!MagQC. 2. The scanned image by different scanners may look different in appearance, while such difference will not and should not affect the pathologists or AI diagnosis. In such we call it “image appearance difference”, which is handled by the image standardization solution in our work. Here what we discussed is the former, i.e. image quality issues which will affect the diagnosis results. Both of above are relevant with the AI model generalization.

First, as we answered review #1 that the image quality for AI diagnosis should have some standard, which is currently missing. It is similar that even in the traditional clinical pathological practices, pathologists have certain requirements for the slides which are suitable for eye-inspection and diagnosis, which is often qualitative and descriptive. And the requirements are often imposed to the step of sample/slides preparation, not scanning. In the digital era, digitalization also introduced another layer of quality, such as out-of-focus and contrast, for example, certainly also including the issues of sample preparation in the traditional practices.

It is tricky to assess how much is the impact of low-quality images in the model training as we may need to consider several different quality issues and what is the portion of "low-quality" data. Our strategy is that we shall build a reliable model using the high-quality data then design generalization solutions and assess if it can tolerate "low-quality" images.

In our current study, the training of the AI model was conducted using only "high quality" WSIs. The QC software only rejects those WSIs that have unacceptable and clear quality issues for pathologists. In reality, pathologists may also request re-scanning of WSIs that are of extremely low quality or even prepare the physical slides again. For testing, the majority of the WSIs used were also of "high quality", although some "low quality" WSIs were included to assess the robustness of the results. In the **sections "Image Quality Assessment Using AI MagQC and Image Appearance Standardization" and " Model Generalization for Invariant Performance Across Multiple Scanners" in Results**, we have demonstrated that WSIs scanned by Scanner C had low quality due to faulty hardware, yet the model performance on these WSIs was consistent with others.

This issue is related to the definition of "high quality" WSI, which may be subjective and may vary among pathologists. As mentioned in your comments, in real-life practice, WSIs may occasionally have tissue folds, out-of-focus scans, staining artifacts, but pathologists are still able to make diagnoses even when the scanning is not 100% perfect. The high quality we defined in this work is different from the definition of pathologists. Of course, the criteria between automated quantitative assessment and human eye qualitative evaluation should be close enough, saying generally in practice hospital pathological departments have around ~1-3% of slides are not suitable for diagnosis. Therefore, ideally, the QC standard should align with pathologists' practice, and WSIs with quality issues that do not affect the pathologists or AI decision should pass the QC test. More systematic efforts are needed to build an association between the qualitative QC done by human pathologists and automated machine QC protocol. Some independent works are conducted with the support of pathologists from different medical centers and we hope to report our results soon in a separated paper.

2. One of the other more unique features of this digital pathology workflow was the creation of the Pathologist-AI Interaction (PAI) utilized as a feedback loop to directly be applied to the model for further training. This is a great idea and can theoretically improve the efficiency of building and maintaining AI algorithms. My one question about this is whether the authors can clarify how the accuracy of these annotations was confirmed? Interobserver variability is a known confounder in prostate cancer

grading, particularly in assessment of Gleason pattern 4. Were multiple pathologists assessing the same WSI and the annotations incorporated into feedback loop if the annotations matched? If different pathologists placed different annotations on the same WSI were they adjudicated in any manner? A continuous feedback loop modifying an AI algorithm is great as long as the accuracy of the annotations are maintained.

Reply: In response to Reviewer 2's question about how the accuracy of the annotations was confirmed, we acknowledge the presence of inter-observer variations in Gleason grading, which can occur in both manual annotations and semi-automatic annotations (i.e., Pathologist-AI Interaction). To ensure high accuracy of the annotations, we took additional steps. In the past few months, we invited 9 pathologists from 5 different centers to review the existing annotations from the manual annotation process and determine the Gleason Score for each WSI in the test set individually.

This multi-annotator review was conducted on the existing fully-manual annotations of prostatectomy specimens, each of which was previously annotated by only one pathologist. However, for the semi-automatic annotations derived from the feedback loop of Pathologist-AI Interaction (PAI), each WSI was assessed by a single pathologist.

3. The authors in this study describe several different pathologists manually annotating images independently for Gleason patterns 3, 4, 5, benign, and stroma on prostatectomy specimens. They also note that the "ground truth" utilized was the Gleason score retrieved from the hospital records, which presumably are the original pathology reports. While the original pathology report for each biopsy specimen would primarily be rendered on a slide-level, and should theoretically match the annotated results on each WSI, the reports for the prostatectomies would assign Gleason grades by incorporating proportions of tumor patterns across many different slides. Therefore, the ground truth for the prostatectomy cases as a whole would not necessarily be expected to match the annotations on individual WSI for these cases. The authors acknowledge this in the Supplemental Materials Section 4 (lines 141-142), but did not mention how they addressed this potential discrepancy. Did they only select WSI from prostatectomy cases with Gleason pattern proportions that matched the original diagnosis? Or did pathologists annotate each WSI in the prostatectomy cases to establish a new ground truth for these cases? It is not clear how many WSI were selected from each prostatectomy or biopsy case, but if a single Gleason score was applied as ground truth across every slide in a prostatectomy case that would seem potentially problematic in terms of assessment of accuracy.

Reply:

We acknowledge the potential challenges associated with using the Gleason score of the entire prostatectomy case as the definitive reference for a single selected slide. Given that a prostatectomy case typically consists of multiple slides, each with its own Gleason score, relying on the aggregate score may introduce inaccuracies. To address this concern, we have made significant revisions in our approach. Instead of relying solely on the Gleason score obtained from the hospital records, i.e., the pathology reports, we adopted a more comprehensive methodology.

Specifically, we sought the expertise of nine pathologists, hailing from five different hospitals, to individually assess the Gleason score for each Whole Slide Image (WSI) of the prostatectomy specimen. This approach was undertaken to mitigate the potential bias or discrepancies that may arise from a single pathologist's assessment. By involving multiple experts, we aimed to incorporate a broader range of perspectives and enhance the robustness of our findings.

To evaluate the concordance between the AI-generated annotations and the Gleason scores determined by the pathologists, we employed the quadratic weighted kappa statistic. This statistical measure takes into account the inter-observer variability inherent in the pathologists' assessments. Please refer to the **"Supplementary Section V: Evaluation of AI model on annotation-level and WSI-level using multiple pathologists' annotations as reference" in the Supplementary** for further details.

4. In prostate biopsies, in addition to benign or outright malignant diagnoses, more indeterminate diagnoses such as "atypical small acinar proliferation", or "high-grade prostatic intraepithelial neoplasia" are rendered not infrequently, particularly in biopsy cases, and have impact on clinical decision-making. Where were these diagnoses categorized? Were they slotted under a "non-malignant" ground truth or "malignant" ground truth? Or were these cases excluded from this study?

Reply: In our study, we focused on annotating "non-malignant" tissues, including Stroma and Benign, as well as Gleason Patterns 3-5. Although HG-PIN and other types of tissues were annotated, the amount was relatively small, and they were not included in the training set. We acknowledge that these categories are important for the diagnosis and prognosis of prostate cancer. In future research, we plan to collect additional annotations for these categories and incorporate them into our AI model. Specifically, we will utilize the incremental learning method to update the existing model such that it will be able to identify more categories, without re-training. Leveraging the "Pathologist-AI Interaction" approach will be an efficient way to collect new annotations and continually update and improve the model.

5. The 3-phase clinical validation experiment involves multiple pathologists assessing 1) glass slides, 2) WSI without AI annotations and then 3) WSI with AI annotations regarding Gleason grading etc. In the Supplemental Section 4 it is noted that 19 slides were examined by each pathologist, with a washout period of at least 20 days. Does this mean the 19 slides examined by each pathologist were the same in all three of these phases? If this is the case, was the order in which the pathologist reviewed these three phases always the same, or did they vary? i.e., did some of them review the WSI with AI annotations first, glass slides second, WSI without AI annotations third, etc? If all 19 slides are identical across all phases of this experiment, and all pathologists reviewed the slides in the same order, even with a washout period of 20 days I wonder if some of the decreased time to render Gleason scoring by the third phase of the study may be a result of recall.

Reply: Yes, the 19 slides reviewed by each pathologist were the same in all three phases, and they reviewed glass slides first, WSIs without annotation second, and

WSIs with annotation first. However, the image IDs were anonymized (renamed randomly) in each phase, and the order of the slides they reviewed was not the same. While this may not be a perfect solution to prevent bias, we believe that the decreased time to render Gleason scoring in the third phase is unlikely to be a result of recall bias.

We acknowledge that the order of phases could potentially affect the results. Therefore, in our future larger-scale multi-center validation, we plan to divide the pathologists into 3 groups, and each group will enter the 3 phases in different orders to mitigate any potential bias.

Overall this was an interesting study with a couple of unique features added to the digital pathology workflow that may have the potential to improve the efficiency of prostate cancer identification and Gleason grading. Clarification of some of the above points may be helpful to put some of the current claims in context.

Reply: Thanks for the comments. We have conducted further analysis and clarified some of the above points. Appreciated your constructive comments!

Reviewers' comments:

Reviewer #1 (Remarks to the Author):

Most major comments addressed and the effort to provide quantitative data for the QC tool and for the scanner variability analysis, as well as slide-level grading metrics is noted and appreciated.

It is a solid body of work, although general concerns about clarity of how all the findings fit together and complexity of figures do remain - defer to authors and editors to try to make things as clear as possible, such as by simplifying figures to have clearer takeaways.

1.4: evaluation metrics - this still doesn't seem to give methods for evaluating the QC or scanner variability (only for Gleason grading) - but the metrics in figures 5 and 6 are mostly self explanatory. Still, could consider briefly mentioning eval metrics for aspects other the Gleason grading since the effort has been made to add this "evaluation metrics" section.

Reviewer #2 (Remarks to the Author):

The authors appear to have addressed most of my questions in their rebuttal letter, and made revisions to their manuscript to address most of the concerns raised in my original review. Having additional pathologists review and confirm annotations, for instance, helps strengthen the accuracy of their model and add credence to their conclusions. Some of the revisions made in the manuscript itself could use a little clarification (ie. Supplemental materials, line 152-153, the sentence that "the filename of the WSI were randomly generated" is less important than pointing out that the order the slides were reviewed was different for each phase for each pathologist, limiting recall bias, for instance). But overall my main concerns have been met. Thank you.

Reviewers' comments:

Reviewer #1 (Remarks to the Author):

Most major comments addressed and the effort to provide quantitative data for the QC tool and for the scanner variability analysis, as well as slide-level grading metrics is noted and appreciated.

It is a solid body of work, although general concerns about clarity of how all the findings fit together and complexity of figures do remain - defer to authors and editors to try to make things as clear as possible, such as by simplifying figures to have clearer takeaways.

1.4: evaluation metrics - this still doesn't seem to give methods for evaluating the QC or scanner variability (only for Gleason grading) - but the metrics in figures 5 and 6 are mostly self explanatory. Still, could consider briefly mentioning eval metrics for aspects other the Gleason grading since the effort has been made to add this "evaluation metrics" section.

Reply:

We highly appreciate your acknowledgment of the our revision.

We have made several improvements to address these issues regarding of your concerns about clarity and figure complexity. Fig 1C and Fig 6 have been simplified and some contents have been moved to Supplementary.

Regarding the Evaluation Metrics, we have expanded the "evaluation metrics" section to include details on the evaluation methods for scanner variability analysis. Please understand that we are preparing another manuscript that focuses on the QC aspect, and comprehensive QC evaluation details will be presented in that manuscript.

Thank you again for your valuable feedback and suggestions.

Reviewer #2 (Remarks to the Author):

The authors appear to have addressed most of my questions in their rebuttal letter, and made revisions to their manuscript to address most of the concerns raised in my original review. Having additional pathologists review and confirm annotations, for instance, helps strengthen the accuracy of their model and add credence to their conclusions. Some of the revisions made in the manuscript itself could use a little clarification (ie. Supplemental materials, line 152-153, the sentence that "the filename of the WSI were randomly generated" is less important than pointing out that the order the slides were reviewed was different for each phase for each pathologist, limiting recall bias, for instance). But overall my main concerns have been met. Thank you.

Reply:

It's great to hear that most of your concerns have been addressed, and we truly appreciate your feedback. We have made minor adjustments in the manuscript to provide the

necessary clarifications, particularly in the Supplementary materials, as highlighted in the example you mentioned. Thank you for your thorough review and constructive feedback.